

# 20m Africa Rice Distribution Map of 2023

Jingling Jiang [1,2,3], Hong Zhang [2,1,3], Ji Ge [1,2,3], Lijun Zuo[1], Lu Xu [1,2], Mingyang Song [1,2,3], Yinhaibin Ding[1,2,3], Yazhe Xie [1,2,3], Wenjiang Huang[1,3]

[1]Key Laboratory of Digital Earth Science, Aerospace Information Research Institute, Chinese Academy of Sciences, Beijing 100094, China;
[2]International Research Center of Big Data for Sustainable Development Goals, Beijing 100049, China
[3]College of Resources and Environment, University of Chinese Academy of Sciences, Beijing 100049, China

*Correspondence to*: Hong Zhang (zhanghong@radi.ac.cn)

**Abstract.** In recent years, the demand for rice in Africa has been growing rapidly, and in order to meet this demand, the rice cultivation area is also expanding rapidly, thus it is of great significance to monitor the rice cultivation in Africa. The spatial and temporal distribution of rice cultivation in Africa is complex, making it difficult to use a climate-based rice identification method, and the existing rice distribution products are all grid based statistical data with low resolution, unable to obtain accurate rice field location and available labels. To address these two difficulties, based on time-series optical and dual-polarisation SAR data, this study proposes a sample set construction method by fast coarse positioning assisted visual interpretation, and a feature importance guided supervised classification combining multiple temporal optical and SAR features to reduce the impact of rice diversity in Africa. Firstly, we use the time-series statistical features of VH data for fast coarse positioning and screening of possible rice areas, and combine multiple auxiliary data for visual interpretation to make sample set; secondly, based on the complementary information in SAR data and optical data, the 20 meter Africa rice distribution map of 2023 was completed by combining the object-oriented segmentation results of temporal optical images and the pixel based classification results of temporal SAR data features after feature selection. The average classification accuracy of the proposed method on the validation set is more than 85%, and the $R^2$ of the linear fit to various existing statistical data is more than 0.9, which proves that the proposed method can achieve the spatial distribution mapping of rice under the complex climatic conditions in a large region, providing crucial data support for rice monitoring and agricultural policy development. The dataset is available at https://doi.org/10.5281/zenodo.13729353 (Jiang, Zhang et al. 2024).

## 1 Introduction

Rice is the staple food for half of the world's population (Kuenzer and Knauer 2013), providing over a quarter of the calories for approximately half of the population (Wu, Zhang et al. 2022), playing an important role in maintaining global food security and also crucial to the economies of many developing countries (Seck, Diagne et al. 2012, Ajala and Gana 2015). In 2021, rice accounted for approximately 8.3% of the world's major crop production (FAO 2023). In Africa, rice accounted for approximately 3.8% of the main crop yield and 4.7% of the global rice production. Despite its current modest



share, the demand for rice in sub-Saharan Africa is increasing at over 6% annually due to population growth, urbanization, and changes in consumer preferences, surpassing any other staple food (Arouna, Fatognon et al. 2021). In order to meet the higher demand for rice, the synchronous growth of local rice production and imports in Africa, and the expansion of rice area rather than the increase in production, are the main driving forces for the increase in domestic production. In the past thirty

years, the cultivated land area has expanded by about 400,000 hectares per year (Yuan, Saito et al. 2024).

In 2023, in order to promote food and nutrition security in Africa, the African Rice Center proposed the 2030 Africa Rice Research and Innovation Strategy (AfricaRice 2023) to transform the rice based agricultural food system, and the rice area in Africa will continue to grow. Meanwhile, rice cultivation and production are important sources of income for a large number of African households (Hussain, Huang et al. 2020). However, rice cultivation in Africa also faces many challenges. Firstly,

Africa is highly susceptible to the impacts of climate change, such as extreme weather events, changes in precipitation patterns, and rising temperatures, which can have a significant impact on agricultural production (Field and Barros 2014, Ogisi and Begho 2023). Land use changes across Africa, particularly urban expansion and deforestation, also influence the distribution of rice cultivation areas (Lambin and Geist 2008, Bren d'Amour, Reitsma et al. 2017). Consequently, it is essential to obtain high-resolution maps of rice spatial distribution in Africa for monitoring the condition of rice cultivation

across the continent.

In recent years, the global crop mapping datasets that include rice in Africa mainly include SPAM2010 (Yu, You et al. 2020), GAEZ+2015 (Global Agro Ecological Zones) (Frolking, Wisser et al. 2020), SPAMAF2017 (International Food Policy Research 2020), and CROPGRIDS (Tang, Nguyen et al. 2023). SPAM2010 and SPAMAF2017 datasets are based on the SPAM model (Spatial Production Allocation Model) developed by the International Food Policy Research Institute (IFPRI),

which utilizes geographic spatial data such as land use types and crop statistical data as inputs to make reasonable estimates of crop distribution within the decomposed units using a cross entropy approach, with a spatial resolution of 5 minite (~10km). GAEZ+2015 utilized the GAEZ model and FAO's crop statistical data to generate grid distribution products for 26 crops, with a spatial resolution of 5 minite (~10km). CROPGRIDS has generated the latest global georeferenced dataset of 173 crops using 26 published grid datasets, with a spatial resolution of 0.05° (~5.5km). The existing datasets have low

resolution and are all grid maps rather than spatial distribution maps. Moreover, these data is generally outdated, making them of limited significance for monitoring rice cultivation in Africa.

Due to the complementarity of information between SAR data and optical remote sensing data, current large-scale rice mapping benefits from multi-source data that combines SAR data and optical remote sensing data as data sources (Han, Zhang et al. 2021, Shen, Pan et al. 2023, Ginting, Rudiyanto et al. 2024). Current rice mapping methods are usually divided

into: 1) Phenology-based classification methods. For example, Qiu (Qiu, Li et al. 2015) utilized the CCVS (the Combined Consideration of Vegetation phenology and Surface water variations) index, constructed using LSWI and EVI during the rice heading and transplanting stages, to map rice in the complex terrain of southern China. Similarly, Zhang (Zhang, Shen et al. 2023) employed the SPRI index, which describes the growth status from the transplanting to maturity stages, to achieve





sample-free mapping of double-cropping rice. These methods do not require sample data but rely heavily on accurate
phenological information. 2) Methods leveraging time-series curve similarity measures, such as DTW (Dynamic Time
Warping) (Guan, Huang et al. 2016) and its improved version TWDTW (Time Weighted Dynamic Time Warping) (Singh,
Rizvi et al. 2021, Tian, Li et al. 2024), requiring only a small number of rice samples to obtain a standard rice growth curve;
3) Supervised classification methods, including various machine learning methods (Wang, Zang et al. 2020, Zhang, Liu et al.
2020, You, Dong et al. 2021) and rapidly developing deep learning methods in recent years (Zhu, Zhao et al. 2021, Sun,
Zhang et al. 2023). These methods offer several advantages for rice mapping. They do not require phenological information,
making them adaptable to different regions and growing conditions. Additionally, they provide high classification accuracy
and robustness when large amounts of labelled sample data are available. This allows for more precise identification and
mapping of rice fields, even in complex landscapes or where other methods struggle. However, the effectiveness of these
approaches depends on the availability and quality of the training data.

The first challenge in mapping rice in Africa lies in the significant temporal and spatial variability of rice cultivation due to
its tropical and subtropical climate, as illustrated in Fig. 1. The data in this figure is derived from the rice calendar product
RiceAtlas (Laborte, Gutierrez et al. 2017) published in 2017, annotating the months when the main and secondary seasons of
rice planting in Africa end and harvest begins. African rice cultivation includes both single and double cropping systems,
with variations in planting times and growth durations across different seasons. This makes it difficult to apply a uniform
phenological description for mapping rice across the entire continent. Notably, large areas of rainfed rice cultivation
(Balasubramanian, Sie et al. 2007) in Africa lack the distinct flooding signals typical of irrigated rice, which are commonly
used in widely adopted rice mapping methods that rely on detecting flooding periods (Guo, Jia et al. 2019, Zhan, Zhu et al.
2021, Wei, Cui et al. 2022). Consequently, phenology-based rice mapping methods are challenging to apply in Africa.
Similarly, DTW-based approaches are difficult to implement due to the variability in rice cropping intensity and phenology,
which hinders the identification of a standard rice growth curve. Therefore, integrating time-series data with supervised
classification emerges as the primary strategy for mapping rice spatial distribution in Africa. However, the main challenge of
this approach lies in constructing sample set. Existing rice distribution products for Africa are grid-based, making it difficult
to quickly identify rice-growing areas for sample set construction. Moreover, the diversity of rice cultivation in Africa—
spanning phenology (including cropping intensity), farming practices (irrigated/rainfed), and environmental conditions
(plains, hills)—complicates the identification of rice fields and makes it challenging to ensure the representativeness and
completeness of the samples.



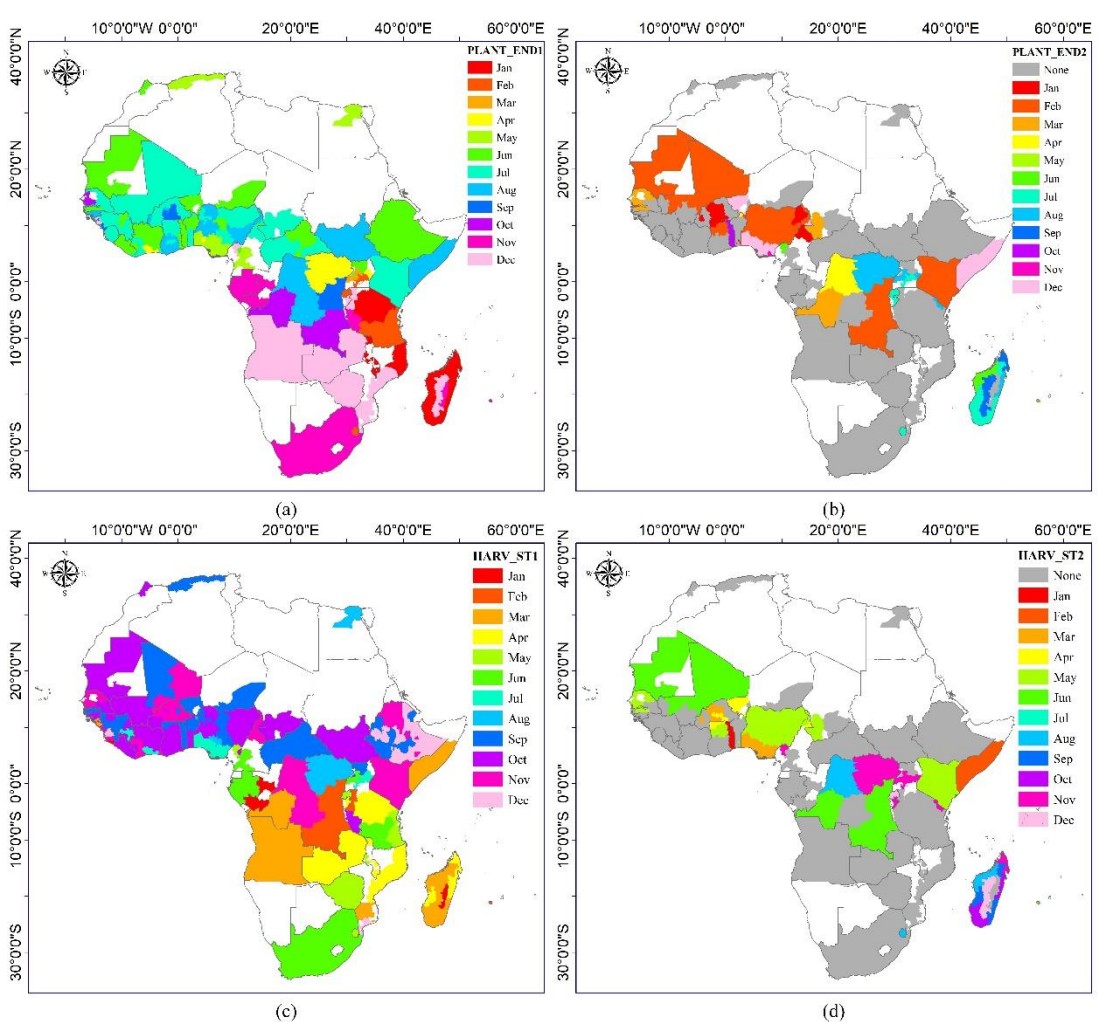

**Figure 1. Rice planting calendar: (a) main rice season planting end date, (b) secondary rice season planting end date, (c) main rice season harvest start date, and (d) secondary rice season harvest start date. Data sourced from RiceAtlas.**

In recent years, the Google Earth Engine (GEE) platform has provided robust support for high-resolution crop mapping. GEE integrates extensive remote sensing data and geographic information system tools, enabling rapid processing and analysis of massive time-series datasets (Gorelick, Hancher et al. 2017). In particular, Sentinel satellite data (Sentinel-1 and Sentinel-2) have been widely applied in crop monitoring and mapping due to their high spatial resolution and frequent temporal coverage (Saad El Imanni, El Harti et al. 2022, Waleed, Mubeen et al. 2022, Luo, Lu et al. 2023, Zoungrana,

Barbouchi et al. 2024). Additionally, the GEE platform supports various supervised classification methods, including Random Forest (RF), Support Vector Machine (SVM), and Classification and Regression Trees (CART) (Liu, Zhai et al. 2020, You, Dong et al. 2021, Avcı, Budak et al. 2023). By integrating multi-source time-series Sentinel data with these supervised classification algorithms available on the GEE platform, it has become feasible to achieve large-scale, high-resolution, and high-accuracy mapping of rice distribution in Africa.



In summary, this study employs a multi-source time series data approach combined with classification algorithms to produce large scale and high-resolution rice distribution maps across in Africa. Specifically, to address the challenge of sample collection, time-series statistical features from Sentinel-1 VH data are used for fast coarse positioning of potential rice-planting areas, followed by visual interpretation using various auxiliary datasets to create reliable samples. During the classification stage, object-based segmentation results derived from Sentinel-2 optical time-series data are integrated with

feature importance guided Random Forest classification results from Sentinel-1 SAR time-series to obtain more precise rice paddy boundaries and reduce noise in heterogeneous landscapes. This approach successfully generated a 20-meter resolution rice distribution map for Africa in 2023. The research could provide scientific support for rice management in Africa, contribute to improving rice yields, ensure food security, and offer data for addressing climate change. The findings are expected to be valuable for policymakers, agricultural scientists, and farmers alike.

**2 Materials**

**2.1 Study site**

In this study, 34 countries with rice harvested areas exceeding 5,000 hectares, as reported by FAO statistics in 2022, were selected as the study regions for rice spatial distribution mapping (FAO 2022), shown in Fig. 2. These include 3 countries in Northern Africa (Egypt, Morocco, Sudan), 15 countries in Western Africa (Benin, Burkina Faso, Côte d'Ivoire, Gambia,

Ghana, Guinea, Guinea-Bissau, Liberia, Mali, Mauritania, Niger, Nigeria, Senegal, Sierra Leone, Togo), 5 countries in Central Africa (Angola, Cameroon, Central African Republic, Chad, Democratic Republic of the Congo), and 11 countries in Eastern Africa (Burundi, Ethiopia, Kenya, Madagascar, Malawi, Mozambique, Rwanda, South Sudan, Uganda, Tanzania, Zambia). The regional division follows the United Nations' Geoscheme (United Nations 2013).



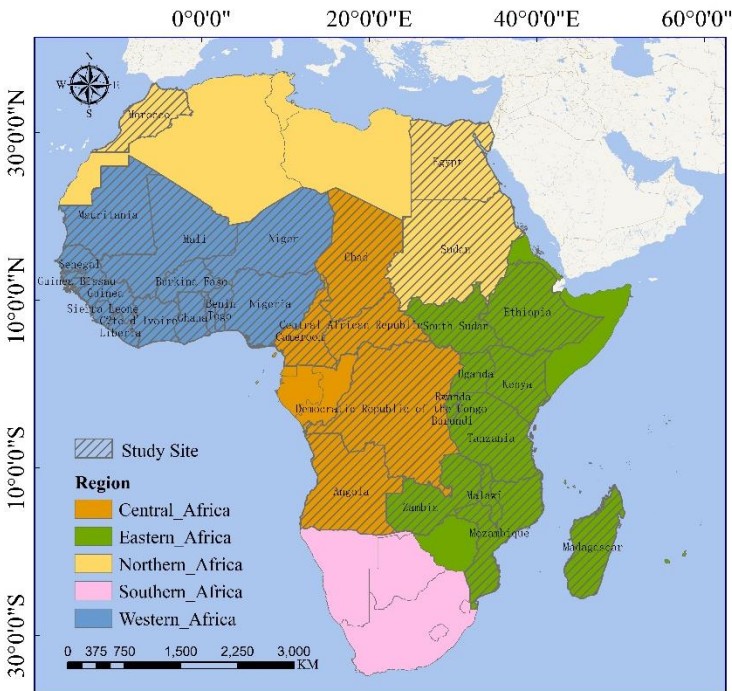

**Figure 2. Study site: 34 countries in Africa with rice harvest areas exceeding 5000 hectares in 2022 according to FAO (diagonally marked area)**

The climatic variations across different sub regions of Africa result in diverse rice cultivation practices. In Northern Africa, dominated by desert and Mediterranean climates, the hot and arid conditions, coupled with scarce rainfall, limit rice cultivation to areas with stable water resources, such as the Nile River basin. Rice is primarily cultivated as a single season crop, relying heavily on irrigation systems. In Western Africa, coastal regions experience tropical rainforest climates, while the interior regions have tropical savanna climates. Rainfall decreases progressively from the coast to inland, leading to rainfed rice cultivation predominantly in coastal areas during the rainy season, which typically spans from May to October, allowing for single-season planting. In inland areas, rice cultivation often depends on flood irrigation or irrigation systems, enabling multi-season cropping. Central Africa also features tropical rainforest and savanna climates, but with uneven rainfall distribution across seasons. As a result, phenological patterns of rainfed rice vary widely in rainforest areas, while rice cultivation in savanna areas partly depends on seasonal flooding or irrigation. In Eastern Africa, the highland regions are characterized by warm and humid mountain climates, where rice cultivation primarily relies on natural rainfall, with the main rainy seasons occurring from April to June and October to December. In contrast, lowland areas have tropical savanna climates, requiring irrigation support for rice cultivation.



## 2.2 Data source

### 2.2.1 Satellite data

The main data sources in the study are time-series SAR data and optical data. Specifically, the monthly average VH and VV data of Sentinel-1 satellite for the whole year of 2023 were obtained as SAR data input on the GEE platform, and the monthly average B3, B4, B8, and B8A band data of Sentinel-2 satellite for the whole year of 2023 were obtained as optical data input. The substantial volume of data, covering multiple spectral and temporal dimensions, enhances the model's capability to detect seasonal variations and accurately map rice fields in diverse agro-ecological zones across Africa. Table 1 presents the number of satellite images utilized for the monthly average composite across each country within the study site. A total of 29,722 Sentinel-1 (S1) images and 387,439 Sentinel-2 (S2) images were employed in the experiment.

Table 1. Number of satellite images used in the study

| Num | Country | S1 image | S2 image | Num | Country | S1 image | S2 image |
|---|---|---|---|---|---|---|---|
| 1 | Angola | 418 | 19765 | 18 | Madagascar | 1106 | 15324 |
| 2 | Benin | 365 | 2142 | 19 | Malawi | 441 | 3008 |
| 3 | Burkina Faso | 486 | 5126 | 20 | Mali | 1400 | 20949 |
| 4 | Burundi | 207 | 1126 | 21 | Mauritania | 1274 | 17083 |
| 5 | Cameroon | 1319 | 8253 | 22 | Morocco | 1448 | 8933 |
| 6 | Central African Republic | 963 | 9542 | 23 | Mozambique | 1877 | 26645 |
| 7 | Chad | 1139 | 19564 | 24 | Niger | 1565 | 18297 |
| 8 | Côte d'Ivoire | 514 | 5575 | 25 | Nigeria | 1677 | 14716 |
| 9 | Democratic Republic of Congo | 3762 | 55967 | 26 | Rwanda | 238 | 917 |
| 10 | Egypt | 1052 | 16529 | 27 | Senegal | 379 | 4192 |
| 11 | Ethiopia | 1625 | 17062 | 28 | Sierra Leone | 213 | 1872 |
| 12 | Gambia | 86 | 791 | 29 | South Sudan | 659 | 9882 |
| 13 | Ghana | 413 | 4407 | 30 | Sudan | 488 | 29213 |
| 14 | Guinea | 515 | 4704 | 31 | Togo | 120 | 1822 |
| 15 | Guinea-Bissau | 142 | 1233 | 32 | Uganda | 639 | 4534 |
| 16 | Kenya | 972 | 8917 | 33 | United Republic of Tanzania | 1427 | 14807 |
| 17 | Liberia | 245 | 2304 | 34 | Zambia | 548 | 12238 |

### 2.2.2 Land cover data

During the sample set construction phase, cropland data from the European Space Agency's ESA WorldCover data for 2020 and 2021 were used as a reference. By integrating land cover data from two consecutive years, the study ensured better

temporal consistency and reliability in sample selection. The use of this land cover data also facilitated the initial separation

of rice and non-rice areas, supporting more precise training and validation in the subsequent classification processes.

### 2.2.3 Rice Grid Data

During the sample set construction phase, rice grid data from the CROPGRIDS (Tang, Nguyen et al. 2023) grid distribution product released in 2023 was used as a reference.

### 2.2.4 Administrative distribution data of rice planting intensity

In the comparison stage with statistical data, the administrative distribution data of rice planting intensity in RiceAtlas product (Laborte, Gutierrez et al. 2017) were used to map the rice paddy area in the mapping results to planting/harvesting area, and then compared with statistical data. The areas without single and double season information were defaulted to planting single season rice. As shown in Fig. 3.

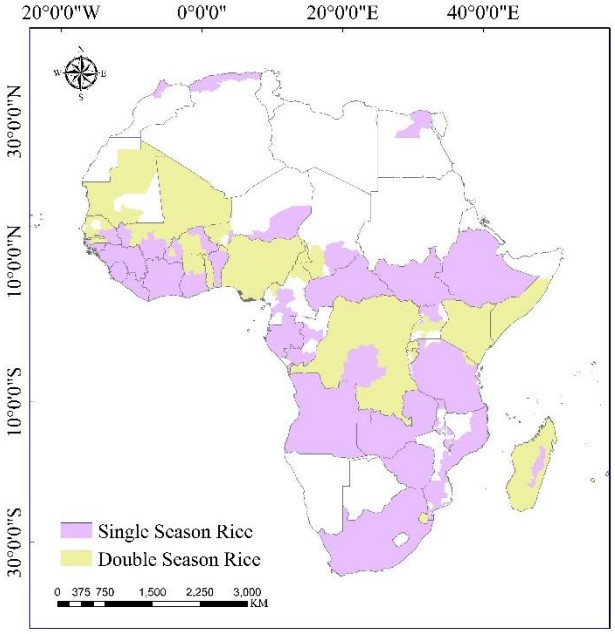

**Figure 3. Administrative distribution map of rice intensity from RiceAtlas**

### 2.2.5 Statistical data

Three kinds of statistical data were used in the study, as shown in Table 2.

Table 2. Statistical data on rice area used in the study

| Statistical Data | Data Time | Retrieve Time |
|---|---|---|
| USDA(United States Department of Agriculture): | 2023 | 2024/02 |



| Rice planting/harvesting area in African countries (USDA 2023) | | |
| --- | --- | --- |
| FAO(Food and Agriculture Organization of the United Nations): Rice harvesting area in African countries (FAO 2022) | 2022 | 2024/03 |
| CARD(COALITION for African Rice Development): Rice planting/harvesting area in CARD countries (CARD 2022) | 2020/2021 | 2024/05 |

**3 Method**

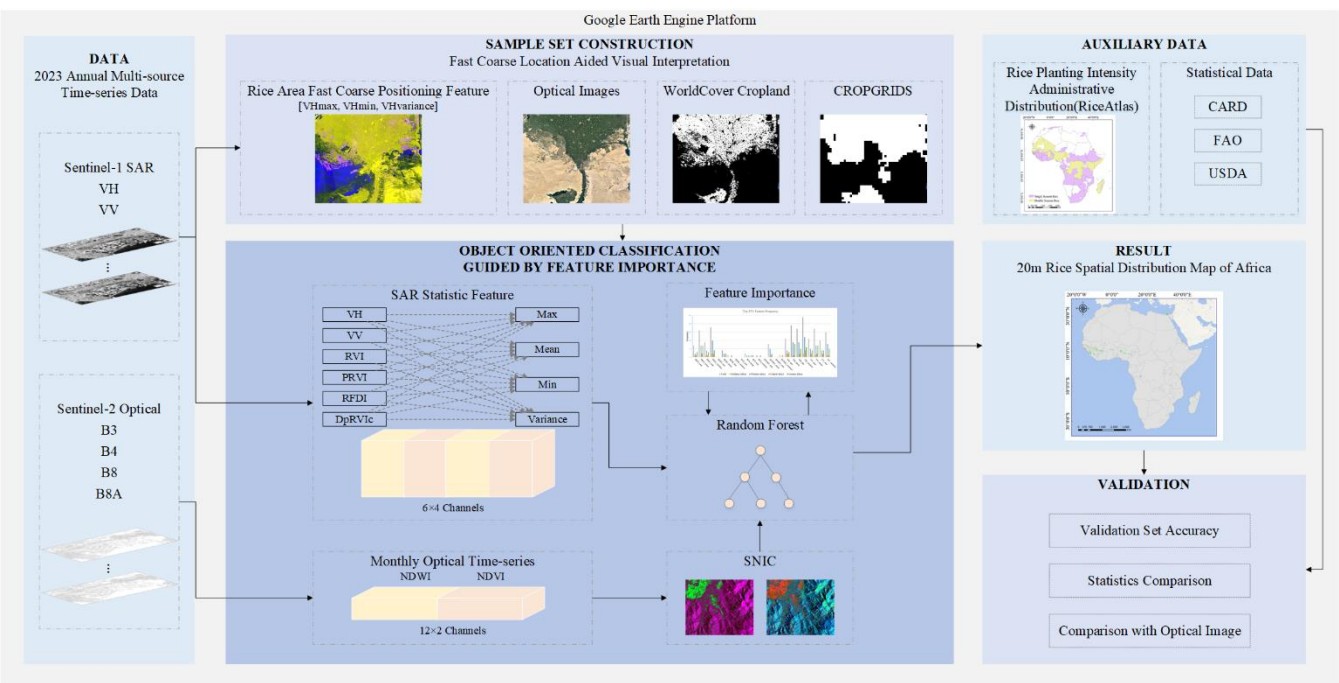

**Figure 4. Flowchart of the proposed rice mapping method (Optical images are from ©GoogleEarth)**

The workflow for mapping the spatial distribution of rice in Africa at a 20-meter resolution is depicted in Fig. 4. The study adopts a multi-source time-series data approach combined with a classifier to achieve large-scale, high-resolution mapping

of rice distribution in Africa. The workflow is primarily divided into two main stages: sample set construction and object-based classification incorporating feature selection.



In the sample set construction phase, VH time-series statistical features are used for the fast coarse positioning of potential rice-growing regions. This is further refined by visually interpreting the samples using ESA WorldCover cropland data, CROPGRIDS rice grid distribution, and optical image.

During the classification phase, object-based segmentation is first performed on optical images to obtain super-pixel results, which helps mitigate the effects of speckle in SAR imagery, enhances classification accuracy, and better captures the complex spatial patterns of rice fields. The mean values of SAR data (VH, VV) and various radar vegetation indices derived from SAR data within these super-pixels are then used as input features. A random forest classifier is applied to train the model, which gives ranks of the importance of the input features. The most important features are selected for a subsequent

classification to produce the rice paddy distribution map. Finally, accuracy validation is conducted using statistical data and validation datasets.

### 3.1 Sample set construction combined with fast coarse positioning

### 3.1.1 Fast coarse positioning of rice planting area

Sun used the statistical features (max, min, variance) of VH time-series data for pseudo-color composite in rice mapping in

Southeast Asia as input features for rice extraction (Sun, Zhang et al. 2023), In this pseudo-color feature map, rice appears purple (VHmin is small, VHmax and VHvariance are large). In the experiment, it was found that rice in Africa also exhibits similar behavior, as shown in Fig. 5. However, wetlands and other land features also exhibit similar characteristics. Therefore, it was only used for fast coarse positioning and preliminary screening of rice regions.

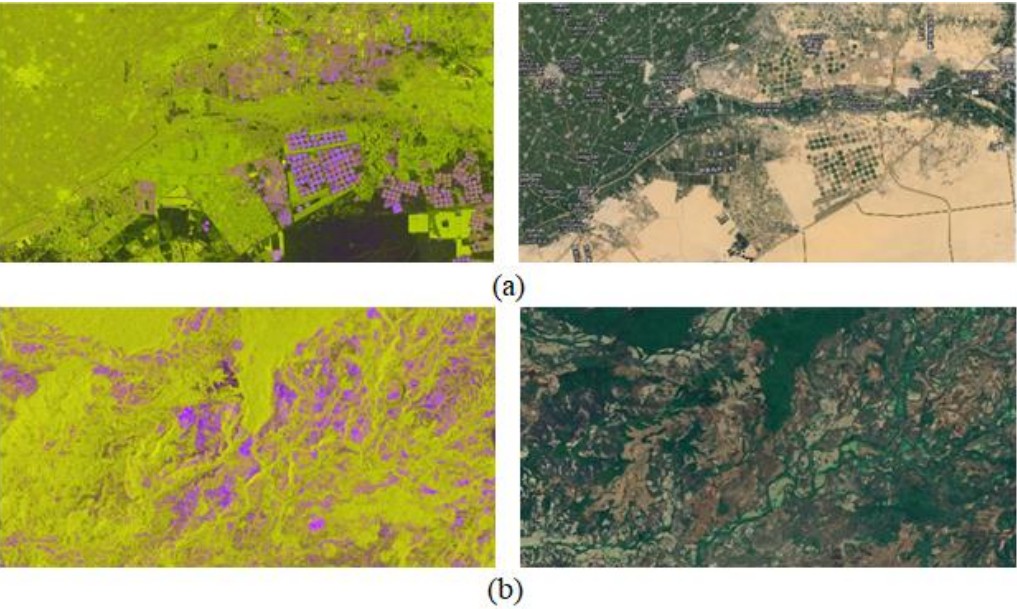

**Figure 5. Pseudo-color composite image (fast coarse positioning feature) and corresponding optical image in Africa (From ©Google Earth) (a) Plain region (b) Hilly region (R: VHmax, G: VHmin, B: VHvariance)**





### 3.1.2 Rice sample set construction

During the experiment, it was found out that wetlands and other land cover types prone to misclassification with rice also appear as purple in the pseudo-color composite image described in Section 3.1.1. Therefore, multiple auxiliary datasets were
used for visual interpretation to make rice sample set. Specifically, after positioning potential rice-plating areas, rice plots were identified and selected as rice samples by cross-referencing the intersections of the rice grid map from CROPGRIDS and cropland distribution maps with corresponding optical imagery. The cropland distribution maps used the union of the cropland classes from WorldCover for the years 2020 and 2021. Additionally, in some countries, existing studies or reports, as listed in Table 3, were consulted.

**Table 3. Reference for rice sample set construction in some countries**

| Country | Reference |
|---------|-----------|
| Benin | (Loko, Gbemavo et al. 2022) |
| Burkina Faso | (Barro, Kassankogno et al. 2021) |
| Egypt | (Mathieu 2022) |
| South Sudan | (FEWSNET 2018) |

In the experiment, 50-300 rice plots were selected for each country, and 2000 rice points were randomly sampled from these plots as positive samples for the classifier input in each country's classification experiment.

### 3.1.3 Negative sample set

In the classification experiments conducted for each country, dozens of plots for each land cover type (non-rice cropland,
built-up areas, water bodies, wetlands, forests, grasslands, etc.) were uniformly selected based on the WorldCover product. For each land cover type, 300 sample points were randomly selected as negative samples for the classifier input.



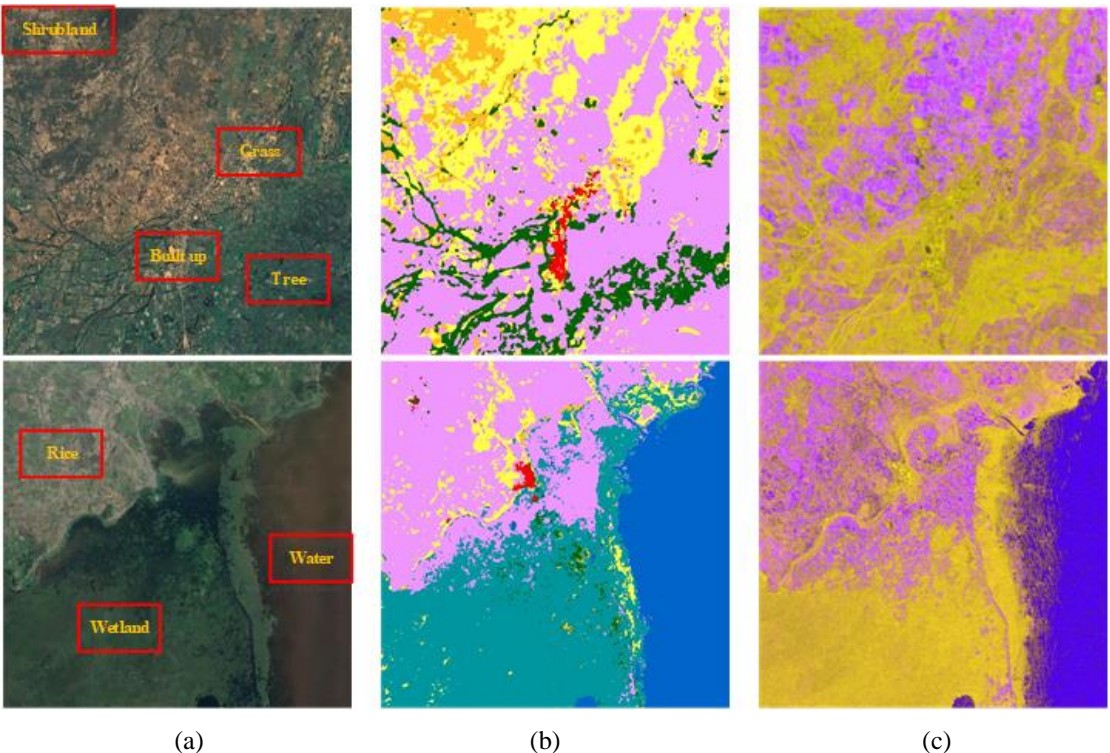

(a)     (b)     (c)

**Figure 6. Example of positive and negative sample regions (a) Optical image(From ©Google Earth) (b) WorldCover2021 from ESA (c) Fast coarse positioning feature**

### 3.1.4 Validation dataset

The validation dataset was constructed similarly to the training sample set. For each country, the validation dataset includes 1,000 rice sample points. Non-rice sample points were uniformly selected based on the number of land cover categories present in the WorldCover product for that country, with 100 sample points chosen for each category (with the cropland category containing only non-rice cropland samples).

### 3.2 Object oriented supervised classification guided by feature importance

### 3.2.1 SNIC Object oriented segmentation

Monthly mean time-series of NDWI and NDVI data from 2023 were used as inputs to perform object-based segmentation using the Simple Non-Iterative Clustering (SNIC) algorithm (Achanta and Susstrunk 2017). This approach was adopted to reduce the fragmentation of rice paddy extraction results and enhance the clarity of rice paddy boundaries. The SNIC algorithm is a super-pixel segmentation method based on the principles of K-means clustering. It initializes seed points on a regular grid as initial cluster centers and assigns each pixel to the nearest cluster based on its distance from the cluster center in both color and spatial dimensions. Since the SNIC algorithm is non-iterative, it requires less computation time and





memory while ensuring connectivity, resulting in good segmentation performance. It is widely used in remote sensing
applications (Tassi and Vizzari 2020, Wang, Meng et al. 2024).

In the experiment, the SNIC algorithm was implemented on the GEE platform with the following parameter settings: seed
distance (size) = 10, segmentation compactness = 0.5, connectivity = 8, and neighbourhood size = 100.

### 3.2.2 Feature importance guided supervised classification

To address the limitations of optical imagery caused by cloud cover in large-scale mapping, SAR features were utilized after
object-based segmentation based on time-series NDVI and NDWI data. The mean values of SAR features within the
segmented super-pixels were used as inputs for supervised classification to achieve more accurate large-scale, high-
resolution rice mapping results. This part of the study employed the Random Forest algorithm available on the GEE platform.
Supervised classification experiments were first conducted for each country, with all SAR data features used as inputs to
determine feature importance rankings. The top-ranked features for each region were then selected, and a second round of
supervised classification was performed using these selected features to obtain the final mapping results.

The SAR features used in the experiment included VH, VV, and four commonly used radar vegetation indices: RVI (Radar
Vegetation Index), PRVI (Polarimetric Radar Vegetation Index), RFDI (Radar Forest Degradation Index), and DpRVIc
(Dual-pol radar vegetation index for GRD data). The statistical features (max, mean, min, variance) for these indices in 2023
were utilized, as defined in Table 4.

**Table 4. Index definition**

| | **Simplified Formula** |
|---|---|
| **RVI** | $\frac{4*\sigma_{HV}}{\sigma_{VV}+\sigma_{HV}}$<br>(Charbonneau, Trudel et al. 2005, Li and Wang 2018) |
| **PRVI** | $\left(1-\frac{\sigma_{VV}}{\sigma_{VH}+\sigma_{VV}}\right)*\sigma_{VH}$<br>(Chang, Shoshany et al. 2018, Sun, Zhang et al. 2024) |
| **RFDI** | $\frac{\sigma_{HH}-\sigma_{HV}}{\sigma_{HH}+\sigma_{HV}}$<br>(Chhabra, Rüdiger et al. 2022) |
| **DpRVIc** | $q*\frac{q+3}{(q+1)^2}, q=\frac{\sigma_{HH}}{\sigma_{HV}}$<br>(Bhogapurapu, Dey et al. 2022) |

### 3.3 Accuracy on validation set

The validation section first performs on the validation set, calculating the user accuracy (UA), producer accuracy (PA), F1-
score, and overall classification accuracy (OA) for rice and non-rice categories:






$$UA = \frac{TP}{TP + FP} \tag{1}$$

$$PA = \frac{TP}{TP + FN} \tag{2}$$

$$F1 = 2 \times \frac{UA \times PA}{UA + PA} \tag{3}$$

$$OA = \frac{TN + TP}{TN + TP + FN + FP} \tag{4}$$

Where TP is true positive, FP is false positive, TN is true negative, and FN is false negative.

## 4 Results


In this section, the results and accuracy will be presented from five aspects: feature screening results, mapping and analysis of rice spatial distribution, comparison of rice area statistics results, validation set accuracy, and comparison of optical images.

### 4.1 Feature importance

Due to the few coverage of SAR images in Angola and Sudan, these two countries only use optical images as classification input features. In the experiments of the remaining 32 countries, a total of 24 statistical features (max, mean, min, and variance) of VH, VV, RVI, PRVI, RFDI, and DpRVIc were input into random forest training to obtain feature importance ranking results. The frequency of each feature in the top 25% of feature importance ranking for each country was calculated according to the UN divided African sub region, as shown in Table 5 and Fig. 7.

**Table 5. Regional statistics on the frequency of features appearing in the top 25% of importance rankings (descending order)**

| Total | | Northern | | Western | | Central | | Eastern | |
|---|---|---|---|---|---|---|---|---|---|
| Feature/Frequency | | Feature/Frequency | | Feature/Frequency | | Feature/Frequency | | Feature/Frequency | |
| VH_variance | 23 | PRVI_variance | 2 | VH_variance | 24 | VH_variance | 4 | VH_mean | 8 |
| VH_mean | 18 | VH_max | 2 | VH_mean | 19 | PRVI_mean | 2 | PRVI_mean | 7 |
| PRVI_variance | 17 | VH_variance | 2 | PRVI_variance | 18 | PRVI_variance | 2 | VV_mean | 7 |
| VH_min | 17 | PRVI_max | 1 | VH_min | 17 | VH_max | 2 | VH_variance | 6 |
| PRVI_mean | 16 | PRVI_min | 1 | VV_mean | 17 | VH_min | 2 | VH_min | 5 |
| VV_mean | 16 | VH_mean | 1 | PRVI_mean | 16 | VV_max | 2 | VV_variance | 5 |
| VV_variance | 15 | VH_min | 1 | VV_variance | 15 | VV_mean | 2 | PRVI_variance | 4 |
| VV_max | 11 | VV_mean | 1 | VH_max | 11 | VV_min | 2 | VV_max | 4 |
| VH_max | 10 | VV_variance | 1 | VV_max | 11 | PRVI_max | 1 | VV_min | 4 |
| VV_min | 10 | PRVI_mean | 0 | VV_min | 10 | PRVI_min | 1 | PRVI_max | 3 |
| PRVI_min | 9 | RFDI_max | 0 | PRVI_min | 9 | DpRVIc_mean | 1 | PRVI_min | 3 |
| DpRVIc_mean | 8 | RFDI_mean | 0 | DpRVIc_mean | 8 | DpRVIc_variance | 1 | VH_max | 3 |
| PRVI_max | 6 | RFDI_min | 0 | PRVI_max | 7 | VH_mean | 1 | RFDI_mean | 2 |
| RFDI_mean | 4 | RFDI_variance | 0 | RFDI_mean | 4 | VV_variance | 1 | DpRVIc_mean | 2 |





| RVI_mean | 2 | RVI_max | 0 | RVI_mean | 2 | RFDI_max | 0 | RFDI_min | 1 |
|---|---|---|---|---|---|---|---|---|---|
| RFDI_min | 1 | RVI_mean | 0 | RFDI_min | 1 | RFDI_mean | 0 | RVI_mean | 1 |
| RVI_min | 1 | RVI_min | 0 | RVI_min | 1 | RFDI_min | 0 | RVI_variance | 1 |
| RVI_variance | 1 | RVI_variance | 0 | RVI_variance | 1 | RFDI_variance | 0 | RFDI_max | 0 |
| DpRVIc_variance | 1 | DpRVIc_max | 0 | DpRVIc_variance | 1 | RVI_max | 0 | RFDI_variance | 0 |
| RFDI_max | 0 | DpRVIc_mean | 0 | RFDI_max | 0 | RVI_mean | 0 | RVI_max | 0 |
| RFDI_variance | 0 | DpRVIc_min | 0 | RFDI_variance | 0 | RVI_min | 0 | RVI_min | 0 |
| RVI_max | 0 | DpRVIc_variance | 0 | RVI_max | 0 | RVI_variance | 0 | DpRVIc_max | 0 |
| DpRVIc_max | 0 | VV_max | 0 | DpRVIc_max | 0 | DpRVIc_max | 0 | DpRVIc_min | 0 |
| DpRVIc_min | 0 | VV_min | 0 | DpRVIc_min | 0 | DpRVIc_min | 0 | DpRVIc_variance | 0 |

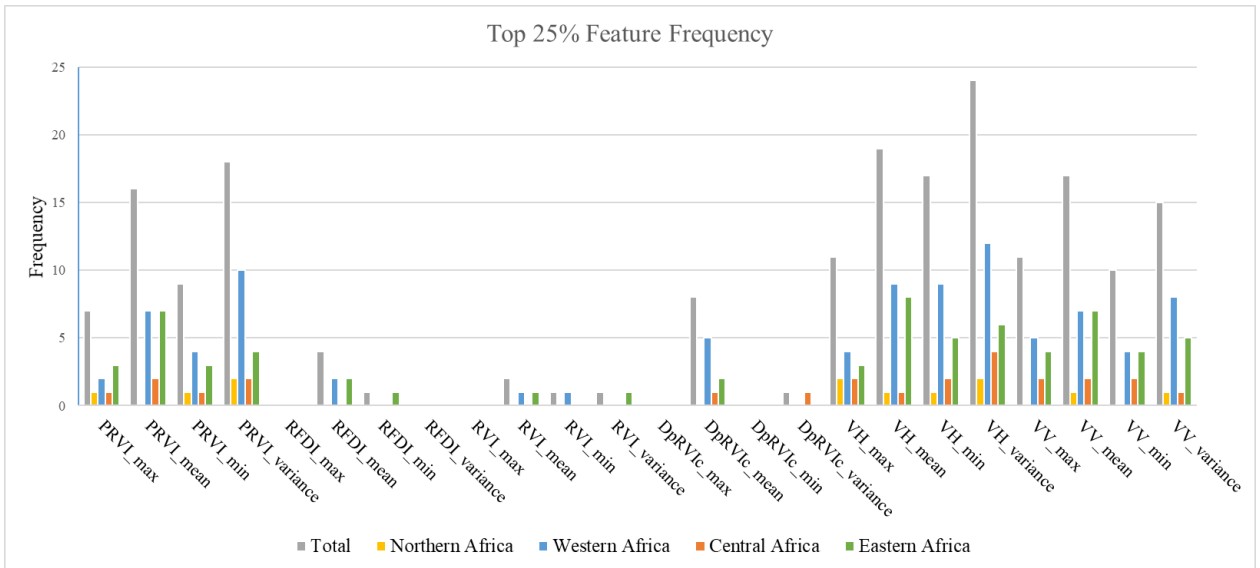

**Figure 7. Regional statistics on the frequency of features appearing in the top 25% of importance rankings (sort by feature)**

In Table 4, the features highlighted in red represent those with the highest frequency within the top 25% of importance rankings for each region (including features with tied frequencies). It can be observed that the top 25% features vary significantly across sub-regions, with the only common feature being VH_variance. Therefore, in the Random Forest supervised classification, each sub-region used the features ranked in the top 25% in frequency for that specific sub-region.

Fig. 8 illustrates an example of selected features, focusing on an area southwest of Lake Alaotra in Madagascar. The classification features used in the supervised classification for this region include six features specific to East Africa: VH_mean, PRVI_mean, VV_mean, VH_variance, VH_min, and VV_variance. These features were combined into two groups for pseudo-color composites, where clear distinctions between rice fields and other land cover types, including wetlands and grasslands that are prone to misclassification, can be observed. This demonstrates that the selected features effectively differentiate rice from other land cover types, enabling accurate spatial mapping of rice distribution. Additionally, the mean values calculated from object-based segmentation of optical imagery improved the representation of SAR image noise and fragmented plots while preserving clear boundaries.





Earth System
Science
Data

**Figure 8. Example of pseudo-color composites using selected time-series SAR features: (a) optical image(From ©Google Earth) (b) pseudo-color composite 1 (R: VH_min, G: VH_variance, B: VH_mean) (c) mean values of pseudo-color composite 1 overlaid on the object-based segmentation result from NDVI time series (d) pseudo-color composite 2 (R: VV_variance, G: VV_mean, B: PRVI_mean); (e) mean values of pseudo-color composite 2 overlaid on the object-based segmentation result from NDVI time series.**


### 4.2 Results of rice spatial distribution mapping

Fig. 9 shows the final 20 meter resolution spatial distribution map of rice across Africa. The green areas represent rice. The map on the right displays the gridded result at a 0.5-degree resolution, with the value in the lower left corner of each grid indicating the rice area, measured in 100 hectares per grid.

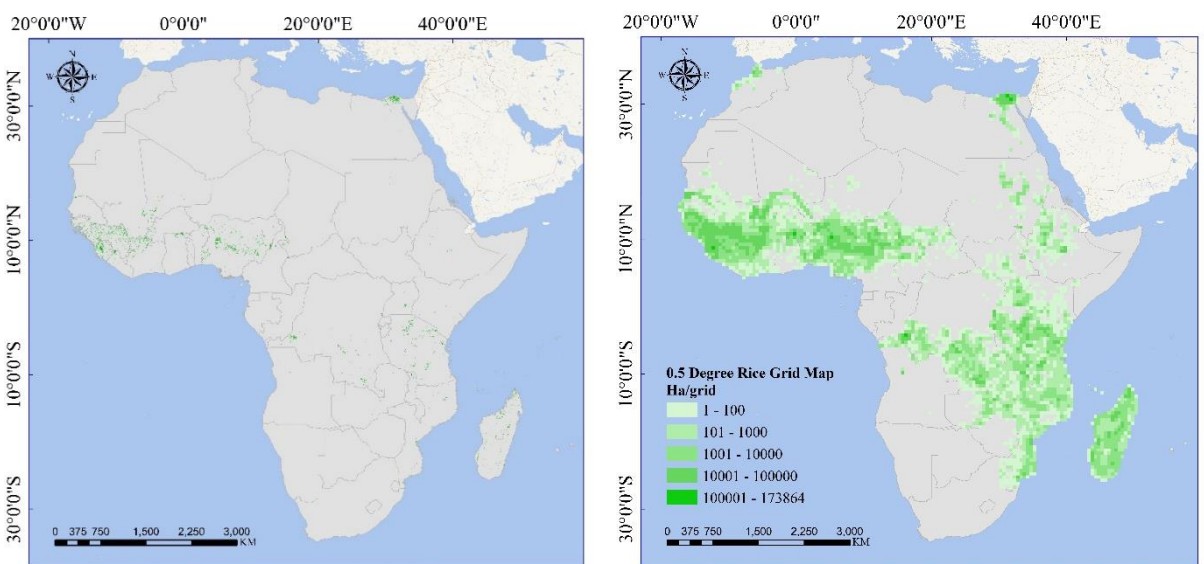


**Figure 9. Rice mapping result in Africa (a) 20 meter spatial distribution map (b) corresponding 0.5° grid map**

**Table 6. Country-level statistics of rice area in Africa based on the 20m spatial distribution map for 2023.**

| Num | Country | Paddy Area/Ha | Single Season Paddy Area/Ha | Double Season Paddy Area/Ha | Planting Area/Ha |
|---|---|---|---|---|---|
| 1 | Angola | 30375 | 30375 | 0 | 30375 |
| 2 | Benin | 149095 | 82340 | 66755 | 215851 |
| 3 | Burkina Faso | 205356 | 137649 | 67707 | 273063 |
| 4 | Burundi | 53626 | 4917 | 48709 | 102335 |
| 5 | Cameroon | 210191 | 17003 | 193188 | 403379 |
| 6 | Central African Republic | 70545 | 70545 | 0 | 70545 |
| 7 | Chad | 283113 | 64938 | 218175 | 501287 |
| 8 | Côte d'Ivoire | 727320 | 727320 | 0 | 727320 |

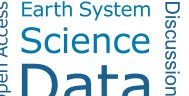

| 9 | Democratic Republic of the Congo | 841988 | 160733 | 681255 | 1523243 |
|---|---|---|---|---|---|
| 10 | Egypt | 689114 | 689114 | 0 | 689114 |
| 11 | Ethiopia | 155157 | 155157 | 0 | 155157 |
| 12 | Gambia | 103316 | 0 | 103316 | 206632 |
| 13 | Ghana | 355311 | 1562 | 353749 | 709060 |
| 14 | Guinea | 1580359 | 1580359 | 0 | 1580359 |
| 15 | Guinea-Bissau | 178277 | 178277 | 0 | 178277 |
| 16 | Kenya | 29610 | 0 | 29610 | 59220 |
| 17 | Liberia | 135214 | 135214 | 0 | 135214 |
| 18 | Madagascar | 865405 | 193680 | 671725 | 1537131 |
| 19 | Malawi | 120866 | 120866 | 0 | 120866 |
| 20 | Mali | 502970 | 91772 | 411198 | 914169 |
| 21 | Mauritania | 63672 | 498 | 63174 | 126846 |
| 22 | Morocco | 40454 | 40454 | 0 | 40454 |
| 23 | Mozambique | 415471 | 415471 | 0 | 415471 |
| 24 | Niger | 45410 | 5246 | 40164 | 85573 |
| 25 | Nigeria | 2446413 | 3157 | 2443256 | 4889668 |
| 26 | Rwanda | 30984 | 0 | 30984 | 61969 |
| 27 | Senegal | 202077 | 19757 | 182320 | 384397 |
| 28 | Sierra Leone | 694314 | 694314 | 0 | 694314 |
| 29 | South Sudan | 48605 | 48605 | 0 | 48605 |
| 30 | Sudan | 52553 | 52553 | 0 | 52553 |
| 31 | Togo | 97076 | 0 | 97076 | 194153 |
| 32 | Uganda | 199103 | 29850 | 169253 | 368356 |
| 33 | United Republic of Tanzania | 1088377 | 1015933 | 72444 | 1160821 |
| 34 | Zambia | 83916 | 83916 | 0 | 83916 |

Table 6 presents the country-level statistics of rice area in Africa based on the 20m spatial distribution map for 2023. The
first column represents the rice paddy area from the 2023 mapping results, the second column shows the single season rice
paddy area calculated based on the rice planting intensity information from RiceAtlas, and the third column represents the
double season rice field area. The fourth column provides the total planting area, with all values reported in hectares. Where

$$\text{Paddy Area} = \text{Single Season Paddy Area} + \text{Double Season Paddy Area} \qquad (5)$$

$$\text{Planting Area} = \text{ Single Season Paddy Area} + 2 * \text{Double Season Paddy Area} \tag{6}$$

The total rice paddy area across Africa in 2023 is approximately 12,795,631 hectares. Among the countries, three have rice areas exceeding 1 million hectares: Nigeria, Guinea, and Tanzania. Six countries fall within the range of 500,000 to 1 million hectares: Madagascar, the Democratic Republic of Congo (DRC), Côte d'Ivoire, Sierra Leone, Egypt, and Mali. Thirteen countries have rice areas between 100,000 and 500,000 hectares: Mozambique, Ghana, Chad, Cameroon, Burkina Faso, Senegal, Uganda, Guinea-Bissau, Ethiopia, Benin, Liberia, Malawi, and Gambia. Lastly, twelve countries have rice areas

between 50,000 and 100,000 hectares: Togo, Zambia, Central African Republic, Mauritania, Burundi, Sudan, South Sudan, Niger, Morocco, Kenya, Rwanda, and Angola. The proportion of rice area by country is illustrated in Fig. 10(a).

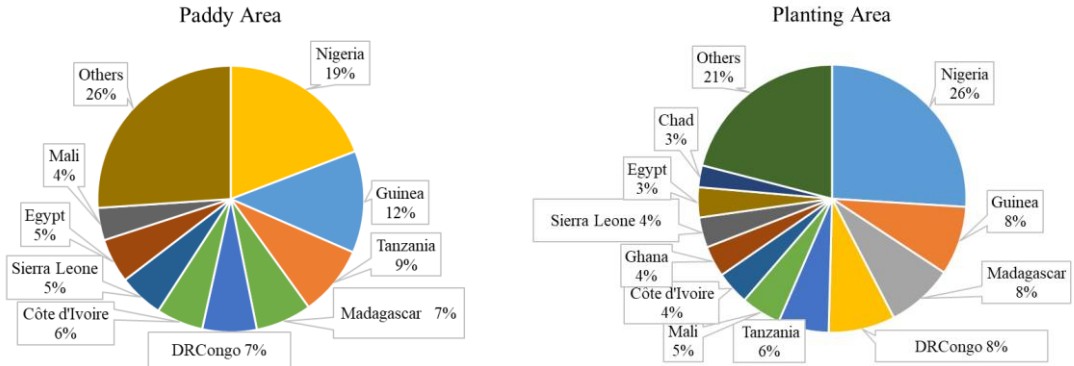

**Figure 10. Proportion of rice area by country in Africa: (a) planting area, (b) paddy area (others: aggregate of countries with areas less than 500,000 hectares).**

In 2023, the total rice planting/harvest area in Africa is approximately 18,739,690 hectares. Five countries have more than 1 million hectares of rice cultivation: Nigeria, Guinea, Madagascar, the Democratic Republic of the Congo (DRCongo), and Tanzania, listed in descending order by area, unless otherwise specified. Six countries have between 500,000 and 1 million hectares: Mali, Côte d'Ivoire, Ghana, Sierra Leone, Egypt, and Chad. Fourteen countries have between 100,000 and 500,000 hectares: Mozambique, Cameroon, Senegal, Uganda, Burkina Faso, Benin, Gambia, Togo, Guinea-Bissau, Ethiopia, Liberia,

Mauritania, Malawi, and Burundi. Nine countries have between 50,000 and 100,000 hectares: Niger, Zambia, Kenya, Central African Republic, Rwanda, Sudan, South Sudan, Morocco, and Angola. The proportion of rice planting area by country is shown in Fig. 10(b).

Regarding single season rice paddy, 12 countries have more than 100,000 hectares: Guinea, Tanzania, Côte d'Ivoire, Sierra Leone, Egypt, Mozambique, Madagascar, Guinea-Bissau, the Democratic Republic of the Congo, Ethiopia, Burkina Faso,

Liberia, and Malawi. For double season rice paddy, 10 countries exceed 100,000 hectares: Nigeria, the Democratic Republic of the Congo, Madagascar, Mali, Ghana, Chad, Cameroon, Senegal, Uganda, and Gambia.



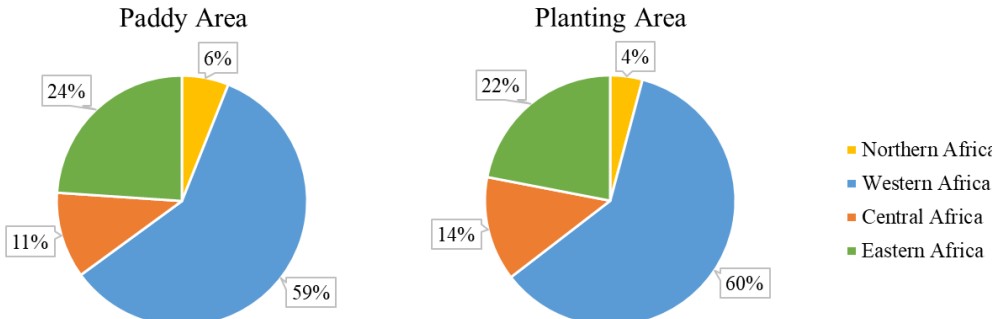

**Figure 11. Proportion of rice area by African sub-region: (a) paddy area, (b) planting area.**

Fig. 11 shows the distribution of rice area by sub-region in Africa. It can be seen that rice planting is primarily concentrated
in Western Africa, followed by Eastern Africa and Central Africa, with the least in Northern Africa. Specifically, all
Northern African countries plant single season rice, covering approximately 800,000 hectares, mainly in Egypt. In Western
Africa, the single season rice area is around 3.7 million hectares, predominantly in Guinea, Sierra Leone, and Côte d'Ivoire,
while the double season area is about 3.8 million hectares, mainly in Nigeria and Mali. In Central Africa, the single season
rice area is approximately 300,000 hectares, and the double season area is about 1.1 million hectares, primarily in the
Democratic Republic of the Congo. In Eastern Africa, the single season rice area is about 2.1 million hectares, mainly in
Tanzania, Mozambique, and Madagascar, while the double season area is around 1 million hectares, primarily in Madagascar
and Uganda. The specific distribution of major production areas is detailed in Table 7.

**Table 7. Distribution of Major Rice-Producing Regions in Africa**

| | |
|---|---|
| **Northern Africa** | |
| Egypt | Predominantly located in the Nile Delta and the Faiyum Oasis. |
| **Western Africa** | |
| Nigeria | Concentrated along the western side of the Kainji Reservoir, as well as along the Niger, Benue, Sokoto, and other rivers and their tributaries. |
| Guinea | Mainly distributed in the coastal plains of the Boffa region in the west, the plains of the Koundara region in the northwest, and along the Niger and Sankarani rivers and their tributaries in the east. |
| Mali | Primarily located along the Niger River and its tributaries in the central and eastern regions. |
| Sierra Leone | Concentrated in the western plains. |
| Côte d'Ivoire | Mainly found along the Bandama River in the northwest, the Bafing region in the west, and the northern areas. |
| **Central Africa** | |





| Democratic Republic of the Congo | Predominantly located near Kinshasa and around Lake Mukamba. |
|---|---|
| **Eastern Africa** | |
| Tanzania | Concentrated in the Mapogoro and Itambaleo regions, the southern areas of Lake Victoria, southern Morogoro, and the Kilimanjaro region. |
| Madagascar | Mainly distributed in the western regions of Lake Alaotra, southwestern areas, and the Ankililoaka region. |

**4.3 Comparison of rice area and statistical data**

Table 8 presents the statistical data of rice planting areas for 34 African countries with more than 5,000 hectares of rice area, listed in alphabetical order. The first column shows the rice planting/harvest area reported by the Coalition for African Rice Development (CARD) for its member countries in 2020/2021. The second column provides the 2022 rice harvest area data from FAO. The third column shows the 2023 rice planting/harvest area reported by USDA. The fourth column presents the 2023 rice planting area derived from this study. All area units are in hectares.

**Table 8. Rice Area Statistics for African Countries**

| Num | Country | 2020/2021 CARD /Ha | 2022 FAO Harvest/ Ha | 2023 USDA/Ha | Planting Area/Ha |
|---|---|---|---|---|---|
| 1 | Angola | 8572 | 8572 | 8000 | 30375 |
| 2 | Benin | 134840 | 134840 | 135000 | 215851 |
| 3 | Burkina Faso | 221052 | 198473 | 220000 | 273063 |
| 4 | Burundi | 50478 | 54441 | 0 | 102335 |
| 5 | Cameroon | 296209 | 156739 | 285000 | 403379 |
| 6 | Central African Republic | 8596 | 36981 | / | 70545 |
| 7 | Chad | 184086 | 177108 | 190000 | 501287 |
| 8 | Côte d'Ivoire | 581766 | 688201 | 730000 | 727320 |
| 9 | Democratic Republic of the Congo | 1442356 | 1888472 | 1660000 | 1523243 |
| 10 | Egypt | / | 646316 | 630000 | 689114 |
| 11 | Ethiopia | 60000 | 60000 | 60000 | 155157 |
| 12 | Gambia | 60097 | 46418 | 65000 | 206632 |
| 13 | Ghana | 414027 | 305000 | 325000 | 709060 |



| 14 | Guinea | 1650217 | 1627939 | 1650000 | 1580359 |
|----|--------|---------|---------|---------|---------|
| 15 | Guinea-Bissau | 126654 | 130291 | 120000 | 178277 |
| 16 | Kenya | 82330 | 29615 | 30000 | 59220 |
| 17 | Liberia | 240000 | 257000 | 240000 | 135214 |
| 18 | Madagascar | 1600000 | 1598207 | 1600000 | 1537131 |
| 19 | Malawi | 76962 | 75787 | / | 120866 |
| 20 | Mali | 874031 | 888116 | 920000 | 914169 |
| 21 | Mauritania | / | 71000 | 75000 | 126846 |
| 22 | Morocco | / | 6320 | 8000 | 40454 |
| 23 | Mozambique | 282000 | 290000 | 290000 | 415471 |
| 24 | Niger | 12566 | 32414 | 30000 | 85573 |
| 25 | Nigeria | 4320100 | 4580000 | 3500000 | 4889668 |
| 26 | Rwanda | 31676 | 32253 | / | 61969 |
| 27 | Senegal | 370750 | 372413 | 370000 | 384397 |
| 28 | Sierra Leone | 944450 | 688549 | 825000 | 694314 |
| 29 | South Sudan | / | 30718 | / | 48605 |
| 30 | Sudan | 8513 | 10753 | / | 52553 |
| 31 | Togo | 98133 | 99958 | 94000 | 194153 |
| 32 | Uganda | 101325 | 260000 | 200000 | 368356 |
| 33 | United Republic of Tanzania | 955729 | 998000 | 1100000 | 1160821 |
| 34 | Zambia | 59601 | 39581 | / | 83916 |

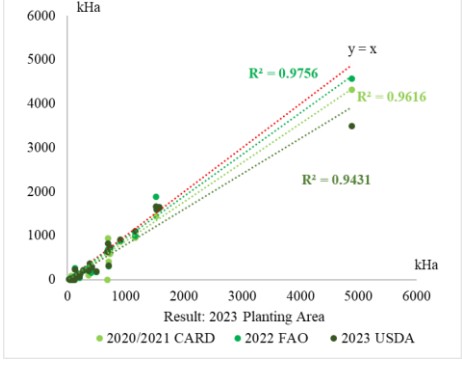
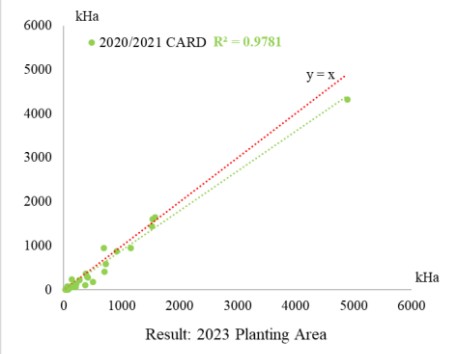
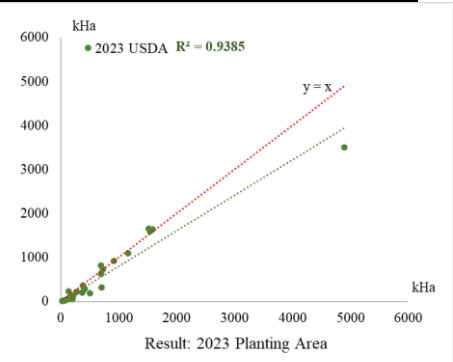



**Figure 12. The linear fitting results between the 2023 rice planting area derived from this study and the existing statistical data, with mapping results as the x-axis and existing statistical data as the y-axis. The red dashed line represents the y = x line. (a) fitting results for all 34 countries, (b) fitting results for 30 countries after excluding those with missing data from the CARD dataset (c) fitting results for 27 countries after excluding those with missing data from the USDA dataset.**

The comparison between the calculated rice planting areas from mapping result and the rice intensity distribution data, alongside existing statistical data, reveals strong linear relationship, as shown in Fig. 12. For all 34 countries, the R² value for fitting with CARD data (2020/2021) is 0.9616, with FAO data (2022) is 0.9756, and with USDA data (2023) is 0.9431. After excluding countries with missing data, the R² for fitting with CARD data (30 countries) improves to 0.9781, while for USDA data (27 countries) it is 0.9385, demonstrating strong consistency.

The figures and tables indicate that in countries with relatively low rice cultivation, the mapped areas generally exceed existing statistical data, shown as points below the y = x line in the fitting plot. In contrast, for countries with larger rice cultivation areas—such as the Democratic Republic of the Congo, Egypt, Guinea, Madagascar, Mali, and Tanzania—the mapped areas closely match existing statistics, with data points near the y = x line. While in Nigeria, the mapped rice cultivation area is significantly higher than existing statistics, represented by points far below the y = x line.

These discrepancies may be attributed to several factors. In developing countries in Africa, data collection and reporting systems are often incomplete and inconsistent, leading to major gaps in the accuracy of reported rice cultivation areas. The issue is further compounded by the dominance of smallholder farming systems, where individual farm sizes are smaller and scattered, making them even harder to track and report on accurately. This often results in underreporting or outdated figures in official statistics. Additionally, rice cultivation in these regions has undergone rapid changes in recent years, with some areas seeing significant increases in planting that aren't being fully captured by traditional reporting methods. Although multiple auxiliary datasets were integrated when constructing rice sample set for this study, the process still heavily relied on expert knowledge. This is particularly challenging in countries with limited rice cultivation, where rice fields are more difficult to identify, leading to sample errors that directly affect mapping accuracy. Moreover, the rice intensity distribution information used to estimate planting areas was published in 2017 and may not fully capture the present situation in 2023, contributing to discrepancies between the mapped data and reported cultivation areas.

### 3.4 Classification accuracy on validation set

The validation results for rice and non-rice classifications across 34 African countries provide a comprehensive insight into the model's performance. The table displays key metrics, including user accuracy (UA), producer accuracy (PA), F1 scores, and overall accuracy (OA). Analyzing these metrics offers an understanding of the spatial variations and classification challenges encountered across different regions.

**Rice Classification Performance:**

**User Accuracy (UA):** The UA for rice classification ranges from 65.26% in South Sudan to 97.51% in Rwanda. The lower values in countries like South Sudan and Niger highlight challenges in correctly identifying rice fields, possibly due to fragmented land use or small cultivation areas.



**Producer Accuracy (PA):** The PA for rice classification spans from 70.78% in South Sudan to 93.17% in Guinea. Higher PA values indicate the model's ability to correctly classify most rice areas, while lower values in regions like South Sudan suggest a tendency for rice areas to be misclassified as non-rice.

**F1 Score:** The F1 scores, combining precision and recall, vary from 67.91% in South Sudan to 94.54% in Guinea. While most countries maintain F1 scores above 80%, lower scores in regions like Angola and Niger highlight difficulties in balancing precision and recall.

**Non-Rice Classification Performance:**

**User Accuracy (UA):** The UA for non-rice ranges from 74.09% in South Sudan to 92.18% in Guinea, with most countries over 85%. High UA values across most countries indicate effective identification of non-rice areas.

**Producer Accuracy (PA):** The PA ranges from 68.92% in South Sudan to 96.55% in Rwanda. Most countries exceed 80%, underscoring consistent performance, though lower values in South Sudan indicate difficulties in distinguishing non-rice areas.

**F1 Score:** The F1 scores for non-rice range from 71.41% in South Sudan to 93.74% in Guinea. Countries with lower scores, such as Niger and Sudan, highlight specific regional challenges in sample set construction with very limited rice cultivation.

**Overall Accuracy (OA):**

The overall accuracy (OA) ranges from 69.76% in South Sudan to 94.17% in Guinea, with a mean of around 86.30%. Countries with extensive rice cultivation, such as Ghana and Senegal, show OAs above 90%, reflecting the model's robustness in regions with more homogeneous and concentrated rice production.

**Key Insights and Implications:**

**Regional Variations:** The variations in accuracy metrics indicate that regional agricultural practices, land use complexity, and data quality play significant roles in model performance. Regions with small, fragmented rice fields or mixed cropping systems, such as South Sudan, Niger, and Angola, present classification challenges that lead to lower accuracy scores.

**Outliers and Challenges:** The box plot (Fig.12) analysis reveals stable and consistent performance across most countries, with median values clustering between 85% and 90%. However, outliers such as South Sudan, Angola, and Niger show lower accuracy scores, suggesting that additional refinement is needed for these regions.

**Model Reliability:** The overall consistency in accuracy metrics across most countries highlights the robustness of the rice mapping methodology. Future improvements could focus on addressing the specific challenges faced in regions with complex agricultural landscapes or limited data availability.

The findings underscore the importance of tailored approaches when applying classification models across diverse African environments. Addressing regional discrepancies will be crucial in enhancing data accuracy and supporting better agricultural policy development across Africa.



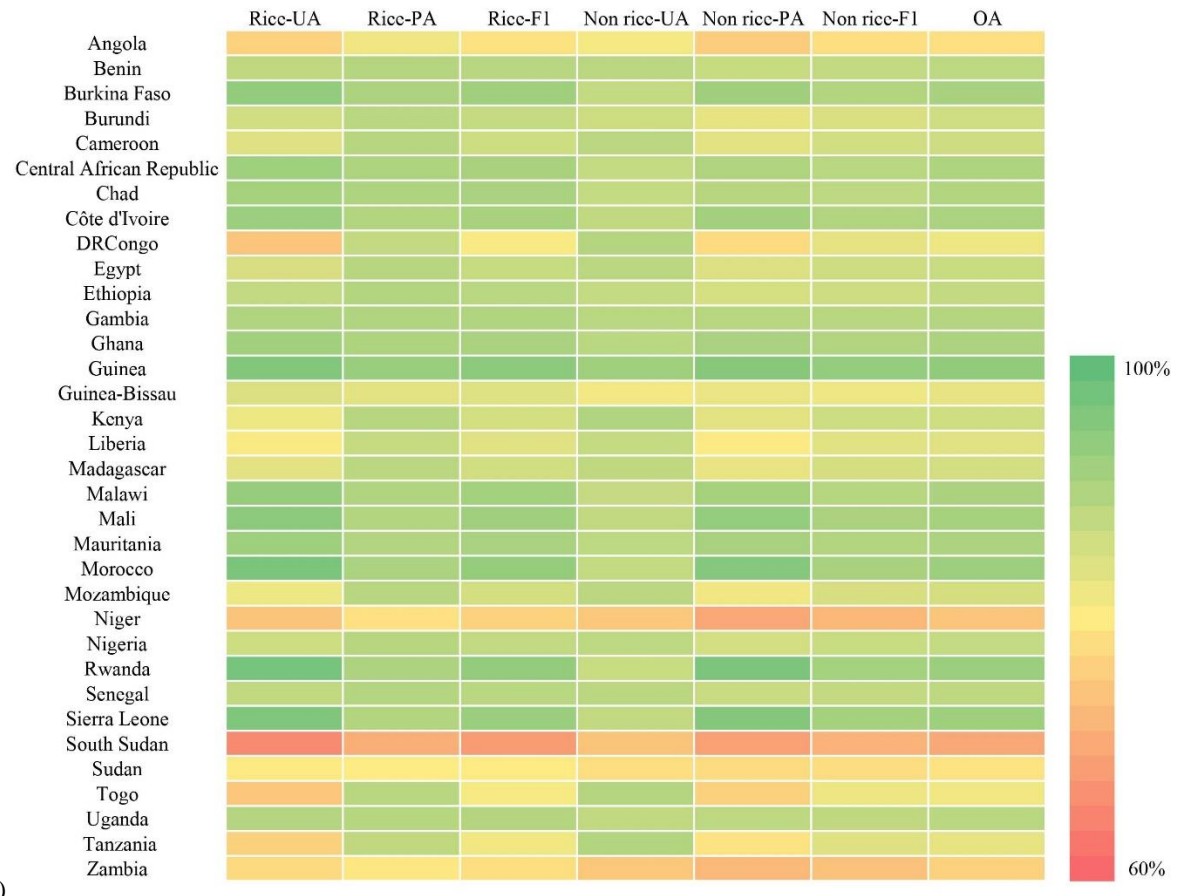

(a)

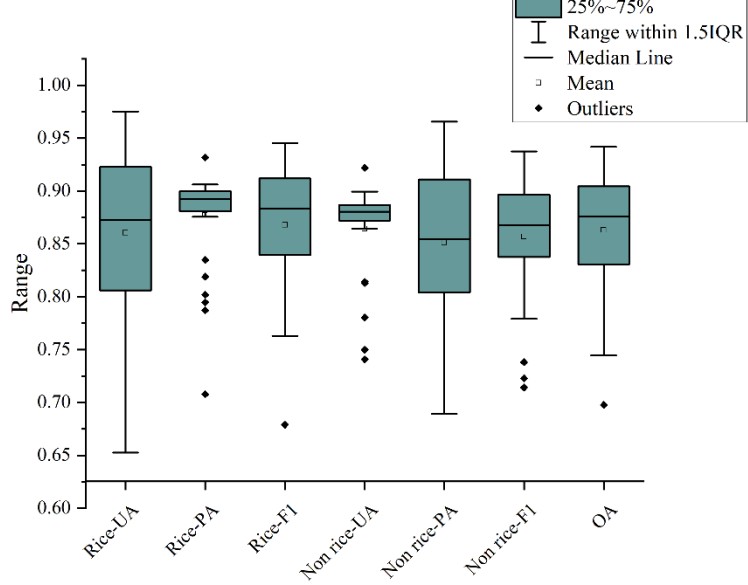

(b)





**Figure 13. Performance on validation set (a) heat map of validation accuracy across 34 African countries (b) corresponding box plot**


## 3.5 Comparison of rice mapping results with optical imagery

Fig. 14 illustrates the comparison between the rice mapping results and corresponding optical image for selected regions in nine major rice-producing countries in Africa (with rice field areas exceeding 500,000 hectares). The examples include both concentrated plantation zones and dispersed smallholder farming areas. The results show a strong alignment between the mapped outputs and the optical images. Additionally, due to the incorporation of the object-based segmentation step, the mapping results exhibit clear boundaries, minimal scattered noise, and fewer misclassifications.


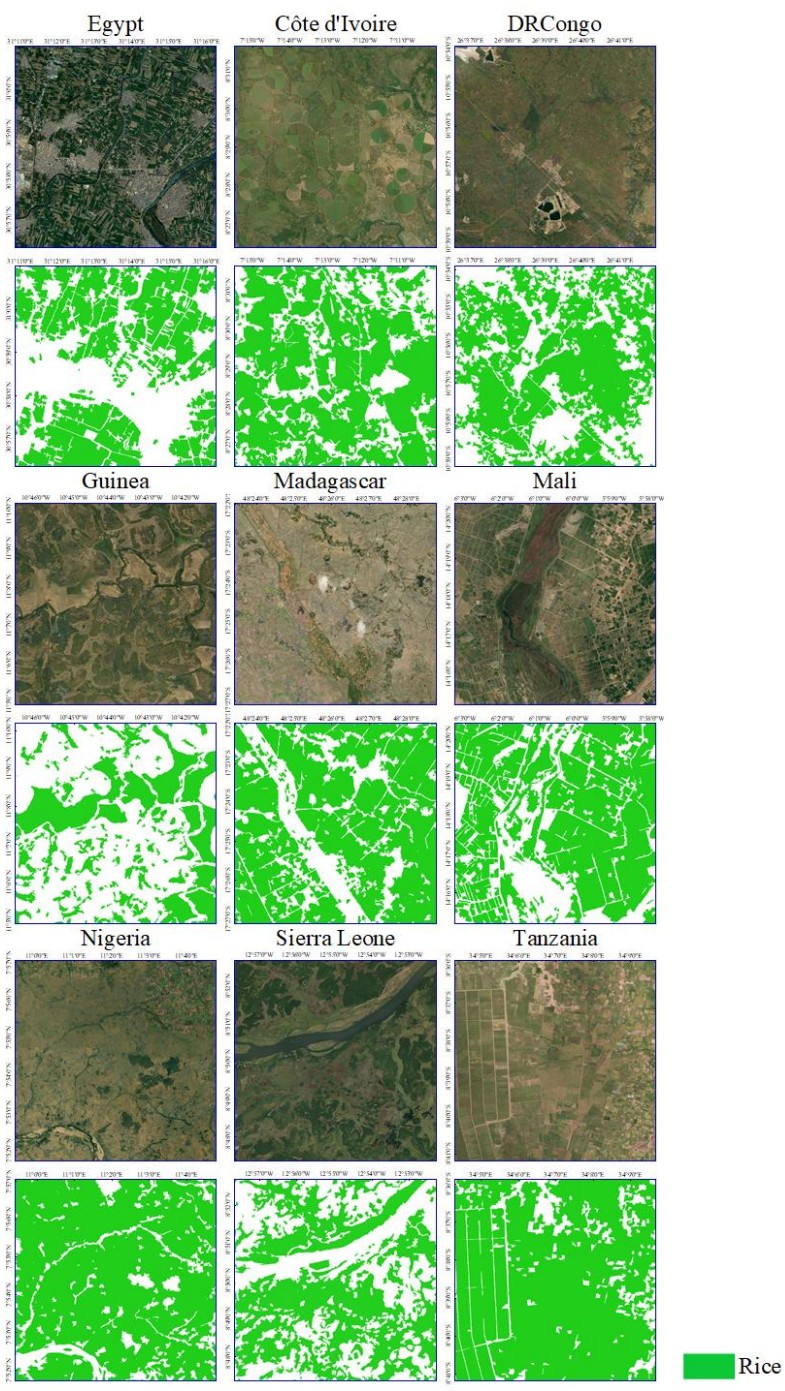

**Figure 14. Examples of rice mapping results and corresponding optical imagery for major rice-producing countries in Africa. For each country, the first row shows the optical imagery (from ©Google Earth), while the second row presents the rice mapping results, with green areas representing rice fields.**



## 5 Discussion

### 5.1 Strengths and limitations

To produce large-scale, high-resolution rice distribution maps across Africa, this study proposed a method effectively combining Sentinel-1 SAR and Sentinel-2 optical imagery, addressing key challenges in sample collection and classification.

By leveraging time-series statistical features from Sentinel-1 VH data for initial fast coarse positioning of potential rice-planting areas and complementing this with visual interpretation using auxiliary datasets, the study efficiently generates reliable samples. During the classification phase, the approach integrates object-based segmentation results from Sentinel-2 optical time-series data with feature importance guided Random Forest classification results from Sentinel-1 SAR time-series data. This combination enhances the precision of rice paddy boundaries and reduces noise in heterogeneous

landscapes.

Despite these strengths, the study acknowledges limitations related to the SNIC algorithm, particularly in the calibration of key parameters—seed distance and neighbourhood size, which affects the size and definition of segmented objects. In this study, it was primarily achieved through a process of trial and visual inspection. While this method provided a practical solution within the context of this research, it lacks the precision and reproducibility necessary for wider application. Future

research should focus on developing more systematic approaches to parameter optimization. This could involve the use of automated tuning algorithms or machine learning techniques that adjust parameters dynamically based on the characteristics of the input data, thereby improving the accuracy, consistency, and scalability of the segmentation process.

Additionally, the study highlights regional variations in the importance of specific features for rice mapping across Africa. Despite these variations, temporal statistical features from SAR data—particularly VH, VV, and PRVI—consistently

demonstrated their utility in capturing the temporal dynamics of rice cultivation. By further exploring and experimenting with these temporal SAR features, future studies could refine rice detection models to be more sensitive to regional differences and temporal changes in Africa. This could involve integrating these features with additional data sources, such as optical imagery or other environmental variables, to create more robust and comprehensive mapping models. Such advancements would not only improve the accuracy of rice mapping in Africa but also contribute to better agricultural

monitoring and decision-making at a broader scale.

### 5.2 Progress and gaps in the National Rice Development Strategy (NRDS) of CARD countries towards 2030 targets

Comparing existing rice planting/harvesting statistics from African countries with the rice planting area results obtained in this study reveals that although rice cultivation in most African countries has fluctuated, there is still a slow upward trend. This aligns with the policy direction of promoting rice cultivation in these countries, though there remains a significant gap

to achieve the 2030 Rice Research and Innovation Strategy for Africa target. Among the countries assessed, 15 have achieved over 80% of the 2030 target, 5 have achieved 60–80%, 7 have achieved 40–60%, and 3 have achieved less than 40%. Of the 9 countries with completion rates below 60%, Tanzania, Senegal, Sierra Leone, and Burkina Faso currently



have substantial rice cultivation areas (greater than 200,000 hectares) but have set high targets. Ethiopia, Liberia, Sudan, Niger, and Kenya have smaller targets but still lag in their current rice cultivation. Countries should develop and adjust their

rice cultivation strategies accordingly to achieve the "Transformation of Rice-based Agri-food Systems for Food and Nutrition Security in Africa" and enhance local food self-sufficiency, ultimately contributing to the SDG goal of zero hunger.

**Table 9. Current rice cultivation areas and 2030 targets for card countries (card 2022), sorted by completion percentage**

| Num | Country | Result | Target/Ha | Ratio | Region |
|---|---|---|---|---|---|
| 1 | Angola | 30375 | 11531 | 263% | Central |
| 2 | Central African Republic | 70545 | 30350 | 232% | Central |
| 3 | Chad | 501287 | 254580 | 197% | Central |
| 4 | Democratic Republic of the Congo | 1523243 | 776000 | 196% | Central |
| 5 | Ghana | 709060 | 372330 | 190% | Western |
| 6 | Burundi | 102335 | 68244 | 150% | Eastern |
| 7 | Malawi | 120866 | 82621 | 146% | Eastern |
| 8 | Uganda | 368356 | 280000 | 132% | Eastern |
| 9 | Cameroon | 403379 | 334764 | 120% | Central |
| 10 | Guinea-Bissau | 178277 | 155046 | 115% | Western |
| 11 | Zambia | 83916 | 80266 | 105% | Eastern |
| 12 | Rwanda | 61969 | 60000 | 103% | Eastern |
| 13 | Togo | 194153 | 193000 | 101% | Western |
| 14 | Benin | 215851 | 242000 | 89% | Western |
| 15 | Gambia | 206632 | 247009 | 84% | Western |
| 16 | Madagascar | 1537131 | 2105690 | 73% | Eastern |
| 17 | Mozambique | 415471 | 570272 | 73% | Eastern |
| 18 | Côte d'Ivoire | 727320 | 1003580 | 72% | Western |
| 19 | Mali | 914169 | 1283970 | 71% | Western |
| 20 | Guinea | 1580359 | 2547881 | 62% | Western |
| 21 | Nigeria | 4889668 | 8523687 | 57% | Western |
| 22 | United Republic of Tanzania | 1160821 | 2200000 | 53% | Eastern |
| 23 | Senegal | 384397 | 775053 | 50% | Western |
| 24 | Ethiopia | 155157 | 327252 | 47% | Eastern |
| 25 | Burkina Faso | 273063 | 627587 | 44% | Western |
| 26 | Sierra Leone | 694314 | 1602103 | 43% | Western |



| 27 | Liberia | 135214 | 326183 | 41% | Western |
|----|---------|--------|--------|-----|---------|
| 28 | Sudan | 52553 | 142856 | 37% | Northern |
| 29 | Niger | 85573 | 252507 | 34% | Western |
| 30 | Kenya | 59220 | 222000 | 27% | Eastern |

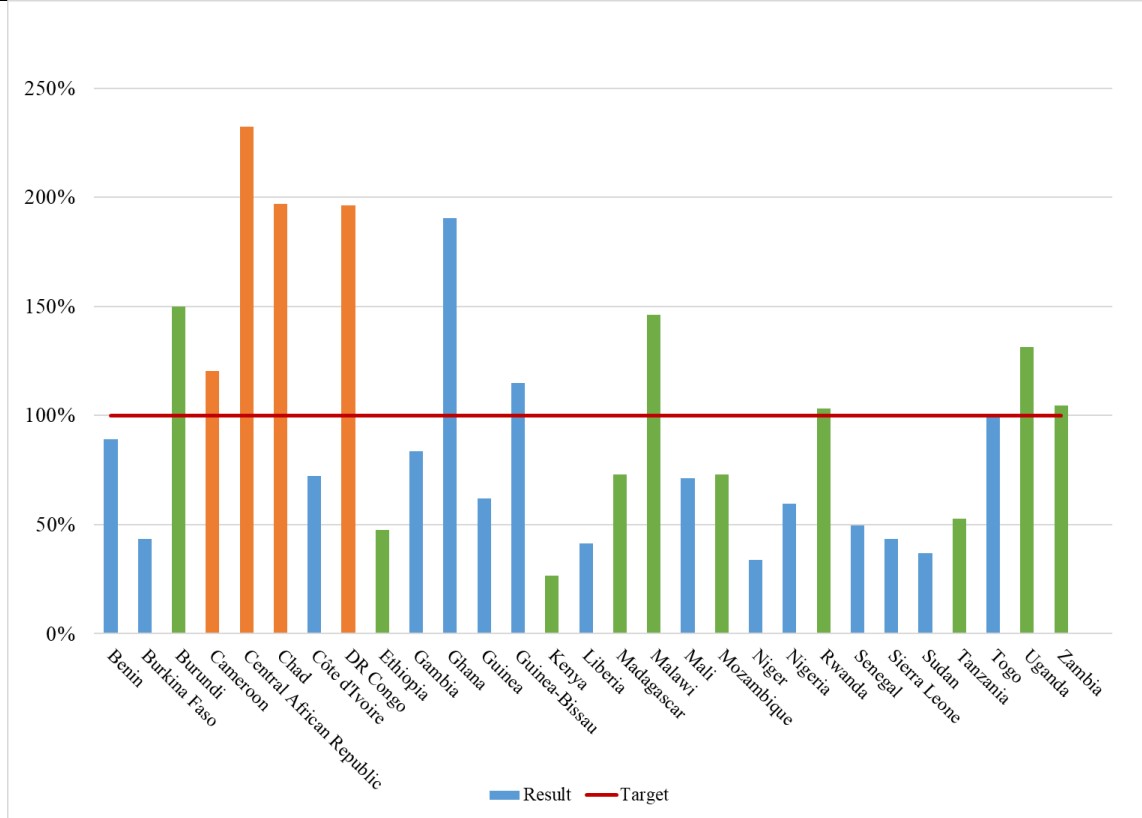

**Figure 15. Comparison of current rice plating areas and 2030 targets for CARD countries**

**6 Data Availability**

The 20m Africa Rice Distribution Map of 2023 can be accessed in the Zenodo data set from the following DOI: https://doi.org/10.5281/zenodo.13729353 (Jiang, Zhang et al. 2024). The spatial reference system of the data set is EPSG:4326(WGS84).

**7 Conclusion**

This study employs temporal SAR data and optical imagery, combined with object-oriented segmentation, and feature importance guided random forest algorithms, to conduct rice extraction experiments in 34 African countries with annual rice





planting areas exceeding 5,000 hectares, achieving 20-meter resolution spatial distribution mapping of rice in Africa for 2023. The average classification accuracy on the validation set exceeded 85%, and the R² values for linear fitting with existing statistical data all surpassed 0.9, demonstrating the effectiveness of the proposed mapping method.

This study marks the first time a high-resolution rice spatial distribution map has been generated for the entire African continent, offering significant advancements in monitoring rice cultivation patterns in the region. The map provides crucial data support for rice yield estimation, climate resilience assessments, and the development of targeted agricultural policies. Moreover, the insights derived from this research can aid in optimizing resource allocation, enhancing food security, and informing decision-making processes for stakeholders ranging from policymakers to local farmers across Africa.


**Author Contributions:** Conceptualization, methodology, software, J.J., M.S. and J.G; validation, formal analysis, J.J. and J.G.; investigation, J.J. and H.Z.; resources, data curation, L.X., Y.D. and Y.X.; writing—original draft preparation, J.J., and H.Z.; writing—review and editing, H.Z., L.X., J.G. and L.Z.; visualization, J.J. ,Y.D., Y.X.; supervision, project administration, H.Z., L.Z., and W.H.. All authors have read and agreed to the published version of the manuscript.

**Funding:** The research was supported by the International Research Centre of Big Data for Sustainable 455 Development Goals (CBAS) [grant numbers CBAS2023SDG001], and the National Key R&D Program of China [grant numbers 2023YFB390620X).

**Acknowledgments:** The authors acknowledge the support of data and computational power provided by the Google Earth Engine platform.

**Conflicts of Interest:** The authors declare no conflict of interest.

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
