# Peer review of "20m Africa Rice Distribution Map of 2023"

_Earth System Science Data, 2024_

## Author Response (AR1)

Dear the reviewers and the editor,

Manuscript ID ESSD-2024-402 entitled "20 m Africa Rice Distribution Map of 2023."

We would like to express our sincere gratitude to the editor and the reviewers for your constructive feedback and thorough review of our manuscript. We have carefully considered all suggestions and have made the corresponding revisions to the manuscript. In addition to addressing the reviewer's comments, we have also refined the overall language to enhance the quality of the paper, and redrawn some of the figures for greater clarity. Below, we provide detailed responses to the reviewer's comments, including clarifications where necessary. We hope these revisions address the concerns and uncertainties raised by the reviewer. In the manuscript and this file, the red parts are revisions suggested by the reviewer 1, blue parts for suggestions of reviewer 2. And the green parts are the changed contents that are intended to improve the expressions.

Sincerely,

Hong Zhang

**Response to Reviewer 1**

Comments to the Author:

This study presents a high-resolution rice distribution map in Africa using an innovative approach that combines time-series optical and SAR data. Given the limitations of current rice distribution products in this region, this study will provide a valuable product for monitoring rice cultivation in African. This product can contribute to assessing food security and the sustainability of rice production in African, such as the evaluation of rice yield and GHGs emission. However, there are several major comments that needed to be addressed:

**RESPONSE:** Thank you very much for your appreciation of our work.**

1. Please provide the full form of all abbreviations when they first appear, such as SAR, to ensure clarity for readers.

**RESPONSE**: Thanks for pointing it out. They are revised in the manuscript.

Line 14 Synthetic Aperture Radar (SAR)

Line 60-61 Land Surface Water Index (LSWI) and Enhanced Vegetation Index (EVI)

Line 116-117 FAO (Food and Agriculture Organization of the United Nations)

Line 144-145 NDWI (Normalized Difference Water Index) and NDVI (Normalized Difference Vegetation Index)

Line 151 European Space Agency's (ESA)

Line 244 GRD (Ground Range Detected) data)

Line 264 UN (United Nations)

Line482 SDGs (Sustainable Development Goals)

2. Section 2.2.1: More details are needed on the criteria used for image screening.

**RESPONSE:** Thanks for your suggestion. Details are added in the manuscript.

Line 140-145

The main data sources in the study are time-series SAR data and optical data for their high temporal and spatial coverage. Specifically, the monthly average VH and VV data of Sentinel-

1 satellite for the whole year of 2023 were obtained as SAR data input on the GEE platform. Because rice is sensitive to NDWI (Normalized Difference Water Index) and NDVI (Normalized Difference Vegetation Index)(De Lima et al., 2021; Zhang et al., 2019), the monthly average B3, B4, B8, and B8A band data of Sentinel-2 satellite for the whole year of 2023 were obtained as optical data input to composite NDWI and NDVI.

3. Section 2.2.4: Is it appropriate to distinguish the distribution of single- and doubleseason rice in 2023 using a crop type dataset in 2017? My main concern is that the planting area of single- and double-season rice in Africa have expanded rapidly in recent years.

**RESPONSE:** Thanks for your comment. This could be a problem as we mention it in 4.3 and sorry for the confusion that it is not well explained in the manuscript. Explanation is added in discussion part.

**Line 450-458:**

Another potential problem is when comparing with statistic data, the administrative distribution data of rice planting intensity in RiceAtlas product is utilized to calculate the planting area from the paddy area of the mapping result. This dataset of year 2017 could lead to gaps between calculated planting area with actual planting area and that with statistical data since rice cultivation expands rapidly in recent years as mentioned in section 4.3. However, there is no up-to-date dataset of rice intensity in Africa. And other datasets including rice intensity in Africa like GCI (Global Cropping Intensity) from year 2001 to 2019 (Liu et al., 2021), and GCI30(Zhang et al., 2021) from year 2016 to 2018, are pixel-level datasets, which are assumed to change more than administrative-level data over time. Therefore, RiceAtlas is chosen as the rice intensity source to balance consistency and data availability. Nevertheless, more up-to-data intensity data can provide more insight into the rice planting status in Africa.

4. Section 3.1: How was the quality of screened samples assessed, and how are these samples distributed?

**RESPONSE:** Thanks for your comment and sorry for the confusion. The "screening" process is part of visual interpretation. The visual interpretation is conducted referring to the intersections of the rice grid map from CROPGRIDS, cropland distribution maps, corresponding optical imagery and the fast coarse positioning feature (R: VHmax, G: VHmin, B: VHvariance), as described in section 3.1.1.

5. The accuracy of the rice distribution map highly depends on image segmentation. Please explain the reason for choosing bands such as B3, B4, B8 and B8A for image segmentation, and provided results demonstrating the image segmentation.

**RESPONSE:** Thanks for your comment. The reason to choose band B3, B4, B8 and B8A is added in the manuscript, as explained in response to comment 2. The effect of image segmentation is presented in Fig. 8, and explained in Line 277-280.

**Line 281-282**

Additionally, the mean values calculated from object-based segmentation of optical imagery improved the representation of SAR image noise and fragmented plots while preserving clear boundaries.

---

## Author Response (AR2)

Dear the reviewers and the editor,

Manuscript ID ESSD-2024-402 entitled "20 m Africa Rice Distribution Map of 2023."

We would like to express our sincere gratitude to the editor and the reviewers for your constructive feedback and thorough review of our manuscript. We have carefully considered all suggestions and have made the corresponding revisions to the manuscript. In addition to addressing the reviewer's comments, we have also refined the overall language to enhance the quality of the paper, and redrawn some of the figures for greater clarity. Below, we provide detailed responses to the reviewer's comments, including clarifications where necessary. We hope these revisions address the concerns and uncertainties raised by the reviewer. In the manuscript and this file, the red parts are revisions suggested by the reviewer 1, blue parts for suggestions of reviewer 2. And the green parts are the changed contents that are intended to improve the expressions.

Sincerely,

Hong Zhang

**Response to Reviewer 1**

**Comments to the Author:**

This study presents a high-resolution rice distribution map in Africa, offering valuable insights into the current state of rice production in the region. It also holds potential for applications such as modeling rice yield and greenhouse gas emissions. However, a major concern lies in the fact that rice cultivation in Africa is heavily constrained by water availability, resulting in two rice cropping systems: irrigated rice and rainfed rice. Notably, the latter accounts for up to 60% of the total rice planting area. Consequently, it is critical to distinguish the distributions of rainfed and irrigated rice, and also elaborate on the specific methods used to distinguish these systems and validate the dataset's accuracy, as accurate identification of rainfed and irrigated rice is critical for enhancing data precision and expanding its application potential.

**RESPONSE:** We appreciate your valuable feedback and thoughtful suggestions. We acknowledge the importance of distinguishing between rainfed and irrigated rice systems, as they are critical for enhancing the applicability of the dataset and important for understanding water use and crop management practices. Currently, the existing rice distribution datasets of Africa are all gridded maps with very limited resolution (highest at 3', ~5.5km), as listed in the table below which is difficult to meet the needs of fine-grained research and decision-making. Therefore, our research focuses on the mapping of rice areas at a high spatial resolution, especially critical in Africa where detailed agricultural data are lacking.

| DATASET                    | DATA TIME | PUBLISHED TIME | RESOLUTION |
|----------------------------|-----------|----------------|------------|
| SPAM2010                   | 2010      | 2020           | 5'         |
| GAEZ+2015                  | 2015      | 2020           | 5'         |
| SPAMAF2017
(Sub Sahara) | 2017      | 2020           | 5'         |
| CROPGRIDS                  | 2023      | 2023           | 3'         |

Table 1 Current rice distribution map of Africa

During the rice sample set construction period of our experiment, we conducted an extensive review of relevant literature, news reports, and various data sources (part of them listed in Table 2) to identify regions with confirmed rice cultivation.

| Country      | Published literature            |
|--------------|---------------------------------|
| Benin        | (Loko et al., 2022)             |
| Burkina Faso | (Barro et al., 2021)            |
| Chad         | (Liang et al., 2017)            |
| Egypt        | (Mathieu, 2022)                 |
| Kenya        | (Menge et al., 2024)            |
| Madagascar   | (Voahanyinirina and Elie, 2007) |

Table 2 References for identifying confirmed rice fields

| Mali         | (Diuk-Wasser et al., 2007)                                           |  |  |
|--------------|----------------------------------------------------------------------|--|--|
| Mozambique   | (Kajisa and Vu, 2023)                                                |  |  |
| South Sudan  | (Fewsnet, 2018)                                                      |  |  |
| Country      | Report                                                               |  |  |
|              | https://zixun.16988.com/news/getDetail?id=666121                     |  |  |
| Angola       | http://www.aape.org.cn/dwgk/zzjg/ywbm/jzgc/gzdt/201801/t20180126_    |  |  |
|              | 6059231.html                                                  |  |  |
|              | https://big5.cctv.com/gate/big5/sannong.cntv.cn/20140408/101454.shtm |  |  |
| Danin        | 1                                                             |  |  |
| Benin        | https://www.yixing.gov.cn/doc/2023/08/31/1166584.shtml               |  |  |
|              | https://www.gov.cn/xinwen/2015-03/23/content_2837484.htm             |  |  |
|              | https://nyncj.taian.gov.cn/art/2022/1/19/art_45390_10292390.html     |  |  |
| Burkina Faso | http://www.chinafarmernet.com/index.php?c=show&id=34948              |  |  |
|              | https://www.imsilkroad.com/news/p/444586.html                        |  |  |
| D            | https://www.yidaiyilu.gov.cn/p/292811.html                           |  |  |
| Burunai      | http://world.people.com.cn/n1/2024/0217/c1002-40178064.html          |  |  |
| Cameroon     | http://shindaya.com/info.php?cid=44&id=1376                          |  |  |
| Com1 in      | https://www.fmprc.gov.cn/web/gjhdq_676201/gj_676203/fz_677316/12     |  |  |
| Gambia       | 06_677632/1206x2_677652/201911/t20191130_8019921.shtml        |  |  |
| Ghana        | https://chinabn.org/invest/t4m/rice_project_background.html          |  |  |
| Guinea       | http://gn.mofcom.gov.cn/article/ddgk/202108/20210803193151.shtml     |  |  |
| C · D        | http://www.focac.org/zfzs/202209/t20220909_10764764.htm              |  |  |
| Oumea-Dissau | http://gw.mofcom.gov.cn/article/jmxw/202312/20231203461473.shtml     |  |  |
| Mauritania   | https://m.investgo.cn/article/gb/fxbg/200306/20030600100287.html     |  |  |
| Morocco      | https://www.changzhou.gov.cn/ns news/278135579454748                 |  |  |

Based on this information, we verified the feasibility of the feature map (R: VHmax, G: VHmin, B: VHvariance) for fast and coarse positioning of potential rice-growing areas, 6 spots of different countries (italic lines in Table 2) are presented in the figure below. The rice fields in Egypt and Mali are irrigated, the rice fields in Mozambique are rainfed, and the other three are mixed or unknown types. In the manuscript, we explain that rice appears purple on the feature map due to a combination of low VHmin values and high VHmax and VHvariance values, which correspond to the flooding period commonly observed in rice cultivation. Although rainfed rice lacks the stable flooding period typical of irrigated rice, it often experiences temporary flooding of varying durations and depths caused by rainfall (Yamamoto et al., 2012; Kwesiga et al., 2019; Panda and Barik, 2021; Mwakyusa et al., 2023). This can be seen in the updated Fig.6, rainfed and irrigated rice region reflect similar purple coloration on the feature map. This means that our method is applicable to both irrigated and rainfed rice and can accurately find potential rice growing areas, but cannot further distinguish between irrigated and rainfed rice.

Current methods using remote sensing data to distinguish irrigated rice and rainfed rice require ground truth data for supervised classification(Vogels et al., 2019), which is not available or

sufficient for the various phenology of rice in Africa. The nearest rice distribution dataset distinguishing irrigated rice and rainfed rice of Africa is SPAMAF2017(International Food Policy Research, 2020), which is modeled using statistical methods and not derived directly from remote sensing data. As a result, its spatial resolution (typically 5 arc-minutes, approximately 10 km) is too coarse to serve as a reference to distinguish irrigated rice and rainfed rice for fine-scale mapping at the 20 m resolution achieved in our study. Moreover, the accuracy of SPAMAF2017 heavily depends on underlying statistical assumptions and input data, which may not fully capture the heterogeneity of rice cropping systems across diverse African landscapes and climate. Additionally, existing irrigation datasets like the FAO's Global Map of Irrigation Areas (GMIA) or the Irrigated cropland of GADAS are also inappropriate to distinguish irrigated and rainfed rice as well for their early date and low resolution, as listed in Table 3. Therefore more in-depth research such as thorough temporal feature analysis is needed in the future to distinguish irrigated and rainfed rice at a fine spatial scale and accommodate with the heterogeneity of cropping systems across diverse African landscapes and climate.

| DATASET            | DATA TIME | RESOLUTION | COVERAGE |  |
|--------------------|-----------|------------|----------|--|
| GMIA               | 2013      | 5'         | Global   |  |
| SPAMAF2017         | 2017      | 5'         | A frico  |  |
| (Sub Sahara)       | 2017      | 5          | Anica    |  |
| GADAS              | 2010      | 0.5° 5'    | Clobal   |  |
| Irrigated cropland | 2019      | 0.3~3      | Giodal   |  |

Table 3 Dataset with irrigation information

Thank you once again for your valuable suggestions regarding this study. In our subsequent work, we will leverage the findings of this study and incorporate more detailed survey data to delve deeper into the distribution patterns of rain-fed and irrigated rice in Africa, thereby providing stronger scientific support for food security and water resource management in the continent.

---

## Author Response (AR4)

Dear the reviewers and the editor,

Manuscript ID ESSD-2024-402 entitled "20 m Africa Rice Distribution Map of 2023." We would like to express our sincere gratitude to the editor and the reviewers for your constructive feedback and thorough review of our manuscript. We have carefully considered all suggestions and have made the corresponding revisions to the manuscript. In addition to addressing the reviewer's comments, we have also refined the captions of certain figures to include copyright statements. Below, we provide detailed responses to the reviewer's comments, including clarifications where necessary. We hope these revisions address the concerns and uncertainties raised by the reviewer. In the manuscript and this file, the red parts are revisions suggested by the reviewer 1. And the green parts are the changed contents responsing to remarks from the preceding review file validation.

Sincerely, Hong Zhang

**Response to Reviewer 1**

Comments to the Author:

The authors have done a good job in addressing the major concerns, except that they still did not improve the low accuracy's of a few countries with relatively smaller area of rice paddies. However, they authors provided reasons why it may not work as well as others. The authors could provide more insights of how to improve that in future maybe.

**RESPONSE:** Thank you very much for your constructive feedback. We greatly appreciate your recognition of our work and your constructive suggestion regarding the improvement of mapping accuracy in countries with relatively small areas of rice paddies. In response to your comment, we have added a discussion to section 5.1 about the possible improvements can be made in the future from two aspects.

To improve the mapping results of the countries with low accuracies, the current mapping results could be used as a reference to expand the sample set for a new round of training and classification. Considering the reduced spatial heterogeneity in these smaller regions compared to the entire African continent, a more detailed analysis of rice phenology could be conducted to enhance classification performance.

To improve the proposed mapping process, we suggest employing weakly supervised learning algorithms to automatically augment the training sample set, guided by feature importance, to ensure the reliability and robustness of the generated samples.

We hope these revisions address your concerns, and we sincerely thank you again for your insightful suggestions, which have helped improve the quality of our manuscript.

Page 28-29, Line 467-477:

Moreover, the sample set in this study was constructed through visual interpretation, assisted by the fast coarse positioning feature. As discussed in the overall accuracy part of Section 4.4, when conducting sample set construction in countries with very small areas of rice (such as South Sudan, Niger, Zambia, Angola, and Sudan), the difficulty to locate rice plots is still huge since wetlands are similar to but much more than rice paddies in the feature map, resulting lower OA in these countries. To enhance the mapping accuracy in such countries, the current mapping results could serve as a reference to develop an expanded sample set for a new round of training and classification in future work. Given the relatively small spatial extent of these regions compared to the entire African continent, the spatial heterogeneity is significantly reduced. This allows for a more detailed analysis of rice phenology, which could substantially improve mapping performance. To improve the effectiveness of the proposed method of mapping rice at a large

scale, weakly supervised learning algorithms could be employed to automatically augment the training sample set and improve classification accuracy in future studies. The sample expansion process could still be guided by feature importance to ensure the reliability and robustness of the generated samples.

**Response to remarks from the preceding review file**

**validation**

To the Author:

Regarding figures 2, 3, 6: I just noticed that your figure ... contains a map. To clarify whether a copyright statement or a credit must be given in the map itself or in the caption, we differentiate between (a) maps entirely created by you, (b) maps created by you but based on layers reused from other originators, or (c) maps simply reused from other originators. An example for (a) is a digital elevation model (DEM) purely based on measurement points collected by you and derived by using a software product. If you use an existing map layer from another originator as a basis for significantly enriching the map with your own content, this would be an example for case (b). Case (c) could be a pure reproduction of Google Maps where your own contribution is rather small (e.g. a city map where you only added a few marks for your study locations). If the map was entirely created by you (case a), there is no need to change the caption or map. Please simply inform us. To the contrary, if your map follows cases (b) or (c), please let us know whether the map is distributed under public domain. If yes, please do not include a copyright statement (copyright is waived) but consider adding a credit to the map or caption. However, if your map follows cases (b) or (c) and is not distributed under public domain, please include at least a credit or even a copyright statement (e.g. © Google Maps), if this is required by the map provider, in the map itself or in the caption.

**RESPONSE:** Thank you for your kind remind and sorry for the trouble. For clarification, GIS country boundaries in Figure 2 are available from GADM (https://gadm.org), which is implemented in the caption. For Figure 3 and 6, we added the copyright statement in the caption.

Page 6, Line 122-124:

Figure 2. Study site: 34 countries in Africa with rice harvest areas exceeding 5000 hectares in 2022 according to FAO (diagonally marked area). GIS country boundaries in Figure 2 are available from GADM (https://gadm.org)

Page 9, Line 166-167